# Macrophage innate training induced by IL-4 and IL-13 activation enhances OXPHOS driven anti-mycobacterial responses

Mimmi LE Lundahl[1,2]*, Morgane Mitermite[3]†, Dylan Gerard Ryan[4,5]†, Sarah Case[6], Niamh C Williams[4], Ming Yang[5], Roisin I Lynch[1], Eimear Lagan[7], Filipa M Lebre[1], Aoife L Gorman[1], Bojan Stojkovic[3], Adrian P Bracken[7], Christian Frezza[5], Frederick J Sheedy[6], Eoin M Scanlan[2], Luke AJ O'Neill[4], Stephen V Gordon[3], Ed C Lavelle[1]*

[1]School of Biochemistry and Immunology, Adjuvant Research Group, Trinity Biomedical Sciences Institute, Trinity College Dublin, Dublin, Ireland; [2]School of Chemistry, Scanlan Research Group, Trinity Biomedical Sciences Institute, Trinity College Dublin, Dublin, Ireland; [3]School of Veterinary Medicine, UCD Veterinary Sciences Centre, University College Dublin, Dublin, Ireland; [4]School of Biochemistry and Immunology, Inflammation Research Group, Trinity Biomedical Sciences Institute, Trinity College Dublin, Dublin, Ireland; [5]Hutchison/MRC Research centre, MRC Cancer Unit, University of Cambridge, Cambridge, United Kingdom; [6]School of Biochemistry and Immunology, Macrophage Homeostasis Group, Trinity Biomedical Sciences Institute, Trinity College Dublin, Dublin, Ireland; [7]School of Genetics and Microbiology, Department of Genetics, Trinity College Dublin, Dublin, Ireland

*For correspondence:
lundahlm@tcd.ie (MLEL);
lavellee@tcd.ie (ECL)

†These authors contributed equally to this work

**Abstract** Macrophages are a highly adaptive population of innate immune cells. Polarization with IFNγ and LPS into the 'classically activated' M1 macrophage enhances pro-inflammatory and microbicidal responses, important for eradicating bacteria such as *Mycobacterium tuberculosis*. By contrast, 'alternatively activated' M2 macrophages, polarized with IL-4, oppose bactericidal mechanisms and allow mycobacterial growth. These activation states are accompanied by distinct metabolic profiles, where M1 macrophages favor near exclusive use of glycolysis, whereas M2 macrophages up-regulate oxidative phosphorylation (OXPHOS). Here, we demonstrate that activation with IL-4 and IL-13 counterintuitively induces protective innate memory against mycobacterial challenge. In human and murine models, prior activation with IL-4/13 enhances pro-inflammatory cytokine secretion in response to a secondary stimulation with mycobacterial ligands. In our murine model, enhanced killing capacity is also demonstrated. Despite this switch in phenotype, IL-4/13 trained murine macrophages do not demonstrate M1-typical metabolism, instead retaining heightened use of OXPHOS. Moreover, inhibition of OXPHOS with oligomycin, 2-deoxy glucose or BPTES all impeded heightened pro-inflammatory cytokine responses from IL-4/13 trained macrophages. Lastly, this work identifies that IL-10 attenuates protective IL-4/13 training, impeding pro-inflammatory and bactericidal mechanisms. In summary, this work provides new and unexpected insight into alternative macrophage activation states in the context of myco-bacterial infection.

## Editor's evaluation

In this valuable study Lundahl et al., examine the role IL-4/13/10 cytokines have on bone marrow-derived macrophages and the modulation of trained immunity – currently, the mechanisms involved in trained immunity remain elusive. The authors demonstrate in a convincing fashion that six days following priming with IL-4/13 was associated with increased oxidative phosphorylation metabolism and enhanced killing of BCG – IL-10 can attenuate this effect. The evidence provided helps support the authors' hypothesis that alternatively activated macrophages help regulate mycobacterial infection. The main weakness in this study concerns epigenetic data to confirm the functional role of trained immunity, this will hopefully be answered by future studies.

## Introduction

A significant feature of macrophages is dynamic plasticity, expressed by their ability to polarize towards distinct activation states (*Lundahl et al., 2017*; *Sica et al., 2015*). Activation with interferon gamma (IFNγ) together with lipopolysaccharides (LPS) yields the bactericidal and pro-inflammatory 'classically activated' or M1 macrophages (*Ferrante and Leibovich, 2012*; *Sica et al., 2015*), whereas activation with the type 2 and regulatory cytokines, interleukin (IL)–4, IL-13, IL-10 and transforming growth factor (TGF)β, results in 'alternatively activated' M2 macrophages, which enhance allergic responses, resolve inflammation and induce tissue remodeling (*Bystrom et al., 2008*; *Mantovani et al., 2004*). Another key difference between these activation states are their metabolic profiles: whereas classically activated macrophages switch to near exclusive use of glycolysis to drive their ATP production, alternatively activated macrophages instead upregulate their oxidative phosphorylation (OXPHOS) machinery (*Van den Bossche et al., 2016*; *Van den Bossche et al., 2017*).

Macrophages are key host innate immune cells for controlling the causative agent of tuberculosis (TB), *Mycobacterium tuberculosis* (*Cohen et al., 2018*). *M. tuberculosis* has arguably caused the most deaths of any pathogen in human history and continues to be the cause of more than a million deaths annually: in 2020, 10 million people fell ill with active TB and 1.5 million died from the disease (*World Health Organization team, 2021*). Whilst classifying macrophages into classical and alternative activation states is a simplification – the reality is a broad spectrum of various differentiation states that is continuously regulated by a myriad of signals (*Sica et al., 2015*) – it is nevertheless considered that for a host to control TB infection, classical macrophage activation is vital (*Jouanguy et al., 1999*; *Philips and Ernst, 2012*). Furthermore, a macrophage metabolic shift to glycolysis has been postulated to be a key for effective killing and thus overall control of TB infection (*Gleeson et al., 2016*; *Huang et al., 2018*). By contrast, alternatively activated macrophages directly oppose bactericidal responses, which has been demonstrated to lead to enhanced bacterial burden and TB pathology (*Moreira-Teixeira et al., 2016*; *Shi et al., 2019*). Due in part to the established ability of classically activated macrophages to kill *M. tuberculosis*, inducing Th1 immunity has been a key aim for TB vaccine development (*Abebe, 2012*; *Andersen and Kaufmann, 2014*; *Ottenhoff et al., 2010*). Apart from targeting adaptive immune memory, another promising approach has emerged in recent years: bolstering innate immune killing capacity by the induction of innate training (*Khader et al., 2019*; *Moorlag et al., 2020*).

Innate training is regarded as a form of immunological memory. It is a phenomenon where a primary challenge, such as a vaccination or an infection, induces epigenetic changes in innate immune cells, which alters their responses following a secondary challenge (*Arts et al., 2018*; *Saeed et al., 2014*; *van der Meer et al., 2015*). Unlike adaptive immune memory, the secondary challenge does not need to be related to the primary challenge. For instance, Bacille Calmette-Guérin (BCG) vaccination has been demonstrated to protect severe combined immunodeficiency (SCID) mice from lethal *Candida albicans* infection; reducing fungal burden and significantly improving survival (*Kleinnijenhuis et al., 2012*). This ability of the BCG vaccine to train innate immunity is now believed to be a mechanism by which it induces its protection against *M. tuberculosis*. Apart from the BCG vaccine, it has been demonstrated that other organisms and compounds can induce innate training, such as β-glucan (*van der Meer et al., 2015*) which induced protective innate training against virulent *M. tuberculosis*, as shown by enhanced mouse survival following in vivo infection, and enhanced human monocyte pro-inflammatory cytokine secretion following ex vivo infection (*Moorlag et al., 2020*).

Conversely, there is a risk that certain immune challenges could lead to innate immune re-programming that hinders protective immune responses. For instance, recent reports have demonstrated how virulent *M. tuberculosis* (*Khan et al., 2020*) and mycobacterial phenolic glycans (*Lundahl et al., 2020*) can program macrophages to attenuate bactericidal responses to subsequent mycobacterial challenge. In the current study, macrophage activation caused by former or concurrent parasitic infections is considered regarding the possibility of innate immune re-programming that hinders protective immune responses against this disease. Geographically, there is extensive overlap between tuberculosis endemic areas and the presence of helminths (*Salgame et al., 2013*). With regard to macrophage activation and combatting tuberculosis, this is an issue as parasites induce type 2 responses, leading to alternative macrophage activation instead of classical activation and associated bactericidal responses (*Chatterjee et al., 2017*; *Li and Zhou, 2013*). Indeed, it was recently demonstrated how products of the helminth *Fasciola hepatica* can train murine macrophages for enhanced secretion of the anti-inflammatory cytokine IL-10 (*Quinn et al., 2019*).

Due to the relevance of macrophage plasticity for the control of mycobacterial infection, we primarily focused upon macrophage activation and training in this context. Unexpectedly, our data indicates that macrophage activation with IL-4 and IL-13 induces innate training that enhances pro-inflammatory and bactericidal responses against mycobacteria. Although murine macrophages trained with IL-4 and IL-13 resemble classically activated macrophages, we identify that they do not adopt their typical metabolic profile, instead retaining heightened OXPHOS activity and notably lacking a dependency on glucose and glycolysis. Lastly, we identify IL-10 as a negative regulator of this innate training response, which may have obscured previous identification of this macrophage phenotype.

## Results
### Prior alternative activation enhances mycobacterial killing

Macrophages are key immune cells for combatting *M. tuberculosis*. Upon infection, alveolar macrophages serve as the initial hosts of the intracellular bacterium (*Cohen et al., 2018*) and bactericidal responses of recruited monocyte-derived macrophages (MDM) are crucial for control and killing of *M. tuberculosis* (*Huang et al., 2018*). Overall, it has been highlighted that an early Th1 driven immune response is key to early eradication of *M. tuberculosis*, where classical activation of macrophages results in effective bactericidal action. However, a complicating factor is the occurrence of concurrent parasitic disease, which instead drives Th2 immunity and alternative macrophage activation, which in turn has been demonstrated to enhance TB pathology (*Moreira-Teixeira et al., 2016*). In this context, we sought to investigate how type 2 responses may induce innate memory, and how such memory could affect macrophage bactericidal properties.

To investigate the effect of IL-4 and IL-13 on macrophage acute and innate memory responses, an in vitro model of BCG infection was used. Murine bone marrow-derived macrophages (BMDMs) were stimulated with IL-4 and IL-13 (M(4/13)) on Day –1, followed by infection with BCG Denmark on either Day 0 or Day 6 (*Figure 1A*). On either day, the macrophages were exposed to roughly 30 bacteria per cell and after three hours extracellular bacteria were removed by washing and internalized bacteria were measured by colony forming units (CFUs), resulting in roughly 1–4 bacteria per cell, depending on the difference in uptake (*Figure 1B*). Internalized bacteria were measured by CFU (*Figure 1—figure supplement 1A*) at 27- and 51 hr post infection and killing capacity was determined as difference in CFU after the 3 hr time point (*Figure 1C*).

Mycobacterial killing was compared to naïve BMDMs incubated with media on day –1 (media control, M(-)) on both Days 0 and 6, and BMDMs classically activated with IFNγ and LPS (M(IFNγ/LPS)) on Day 0. M(IFNγ/LPS) could not be investigated on Day 6 due to reduced viability: by Day 6, M(IFNγ/LPS) displayed reduced cell numbers and upon the addition of live bacteria, all the remaining cells died within 24 hr (not shown). On Day 0, M(IFNγ/LPS) displayed significantly enhanced mycobacterial killing at 27- and 51 hr post infection, whereas M(4/13) displayed comparable killing capacity to naive M(-) (*Figure 1C*). Secretion of TNFα and IL-10 were also measured at 3-, 27-, and 51 hr post infection on Day 0 (*Figure 1—figure supplement 1B*) and Day 6 (*Figure 1D*). These cytokines were chosen as TNFα is critical in the early host response against *M. tuberculosis* (*Bourigault et al., 2013*; *Keane et al., 2001*), while IL-10 can prevent phagolysosome maturation in human macrophages (*O'Leary et al., 2011*) and promote disease progression in mice (*Beamer et al., 2008*). Following BCG infection

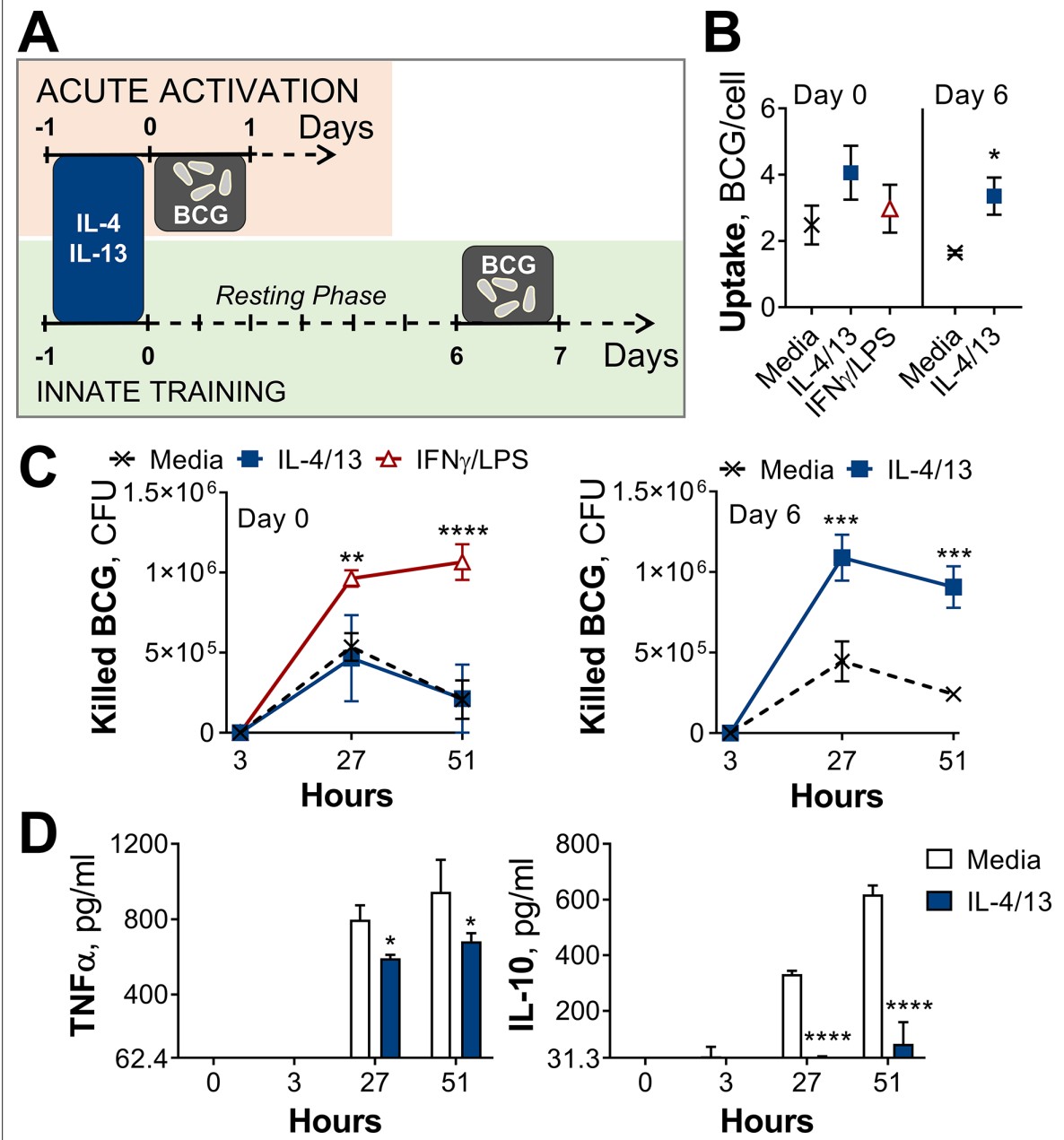

**Figure 1.** Training with IL-4 and IL-13 enhances BMDM mycobacterial killing capacity. (**A**) Schematic of protocol for BCG infection following acute activation (Day 0) or training (Day 6) with IL-4 and IL-13. (**B–C**) BCG Denmark uptake (**B**) and killing after h as indicated (**C**) measured by difference in CFU after 3 hr per $0.5×10^6$ BMDMs on Day 0 or Day 6. BMDMs were incubated with media, IL-4 with IL-13 or IFNγ with LPS for 24 hr on Day –1. (**D**) Secretion of indicated cytokines from BMDMs treated as in (**C**) standardized to $0.5×10^6$ BMDMs. (**B–D**) Representative results (n=2 out of ≥3) showing mean ± SEM (**B–C**) or ± SD (**D**) and analyzed by student's t-test (**B**) or multiple t-tests, with Holm-Sidak correction (**C–D**) compared with media control. * p≤0.05, ** p≤0.01, *** p≤0.001, **** p≤0.0001.

The online version of this article includes the following figure supplement(s) for figure 1:

**Figure supplement 1.** CFU counts and cytokine secretion from BCG killing experiments.

on Day 0 (***Figure 1—figure supplement 1B***), no IL-10 secretion was detected, and only bactericidal M(IFNγ/LPS) secreted detectable levels of TNFα. Consistent with previous work, activation with IL-4 and IL-13 on the other hand did not enhance BCG killing nor induce inflammatory cytokine secretion.

By contrast, on Day 6 (innate training responses), M(4/13) exhibited both heightened mycobacterial uptake (***Figure 1B***) and significantly greater killing capacity compared with untrained M(-), both

at 27- and 51 hr post infection (*Figure 1C* and *Figure 1—figure supplement 1A*). Furthermore, this was accompanied by a near complete abrogation of IL-10 secretion, along with a minor decrease of TNFα compared with M(-) (*Figure 1D*), displaying an overall shift towards a more pro-inflammatory response profile. To investigate the change in phenotype of M(4/13) between the two days tested, a more detailed characterization was carried out.

## Innate training with IL-4 and IL-13 promotes pro-inflammatory responses

The current dogma suggests that classical (M1) macrophage activation induces upregulation of antigen presentation, enhanced secretion of pro-inflammatory cytokines including interleukin (IL)–1β, IL-6, and TNFα, as well as the production of reactive oxygen and nitrogen species (ROS and RNS, respectively) (*Bystrom et al., 2008*; *Sica et al., 2015*). By contrast, alternatively activated macrophages (M2) dampen inflammation, promote angiogenesis and scavenge debris (*Gordon and Martinez, 2010*; *Sica et al., 2015*). These macrophages are identified by their upregulation of chitinase-like 3–1 (Chil3), found in inflammatory zone-1 (Fizz1, Retnla) and arginase (Arg1) (*Gabrilovich et al., 2012*; *Murray and Wynn, 2011*), as well as surface expression of lectins, such as the macrophage mannose receptor (MR, CD206) (*Brown and Crocker, 2016*). To confirm the activation states induced in the present study tallied with the current literature and explore how these activation states may change over time, characterization of activated macrophages was carried out following acute activation and 1 week later (*Figure 2* and *Figure 2—figure supplement 1*). Flow cytometry and qPCR analysis of M(IFNγ/LPS) and M(4/13), compared with inactivated M(-), showed that these activation phenotypes tallied with the literature: M(IFNγ/LPS) had elevated expression of CD80, major histocompatibility complex class II (MHC II) and inducible nitric oxide synthase (iNOS, *Nos2*) – responsible for production of the reactive nitrogen species, nitric oxide (NO) – whereas M(4/13) exhibited heightened expression of CD206, MHC II, *Arg1*, *Chil3,* and *Retnla* (*Figure 2—figure supplement 1A-C*).

To examine their respective cytokine response profiles, M(IFNγ/LPS) and M(4/13) were stimulated with killed *M. tuberculosis* strain H37Rv (hereafter referred to as Mtb) or the TLR1/2 ligand tripalmitoyl-*S*-glyceryl-cysteine (PAM3CSK4) on either Day 0 or Day 6. Acutely activated M(IFNγ/LPS) demonstrated elevated secretion of TNFα, IL-6 and IL-10 in response to Mtb (*Figure 2—figure supplement 1D*) and elevated TNFα and IL-6 in response to PAM3CSK4 (*Figure 2—figure supplement 2A*), compared with M(-). By contrast, M(4/13) had attenuated secretion of each cytokine compared with M(-), following either Mtb or PAM3CSK4 stimulation. However, on Day 6 (innate memory responses), both M(IFNγ/LPS) and M(4/13) secreted greater pro-inflammatory TNFα and reduced anti-inflammatory IL-10 in response to Mtb (*Figure 2A*), and secreted elevated TNFα following PAM3CSK4 stimulation (*Figure 2—figure supplement 2B*). M(4/13) additionally exhibited increased IL-6 secretion in response to both stimuli. As with the BCG infection, M(IL-4/13) responses changed markedly between Day 0 and Day 6, shifting towards a pro-inflammatory response profile which was similar to classically activated macrophages.

To consider whether the shift in M(4/13) to a pro-inflammatory cytokine profile was dependent on epigenetic changes, the DNA methylation inhibitor 5'-deoxy-5'-(methylthio)adenosine (MTA), was employed. This inhibitor has been demonstrated to impede training by BCG (*Kleinnijenhuis et al., 2012*) and β-glucan (*Quintin et al., 2012*) – which promote pro-inflammatory responses – as well as training induced by helminth *Fasciola hepatica* total extract (*Quinn et al., 2019*), which enhances anti-inflammatory responses such as IL-10 and IL-1Ra secretion. The addition of MTA prior to activation with IL-4 and IL-13 on day –1 reduced TNFα, IL-6 and IL-10 secretion induced by either Mtb or PAM3CSK4 on Day 6 (*Figure 2—figure supplement 3A*), resulting in a profile reminiscent of acutely activated M(4/13) (*Figure 2—figure supplement 1D*). By contrast, inhibition of DNA methylation in M(-) resulted in comparable or even increased cytokine secretion. This suggested that DNA methylation following IL-4 and IL-13 activation contributed to the innate training and subsequent enhancement of pro-inflammatory responses. Moreover, in an attempt to identify which type of histone methylation was occurring, the following global histone H3 modifications were investigated by western blot: H3K4me3, H3K27me3, and H3K9me2 (*Figure 2—figure supplement 3B-D*). Yeast β-glucan was used a positive control for innate training; however, none of the tested conditions resulted in significant changes in global methylation at any of these sites, demonstrating a need to investigate specific promoter regions in future studies.

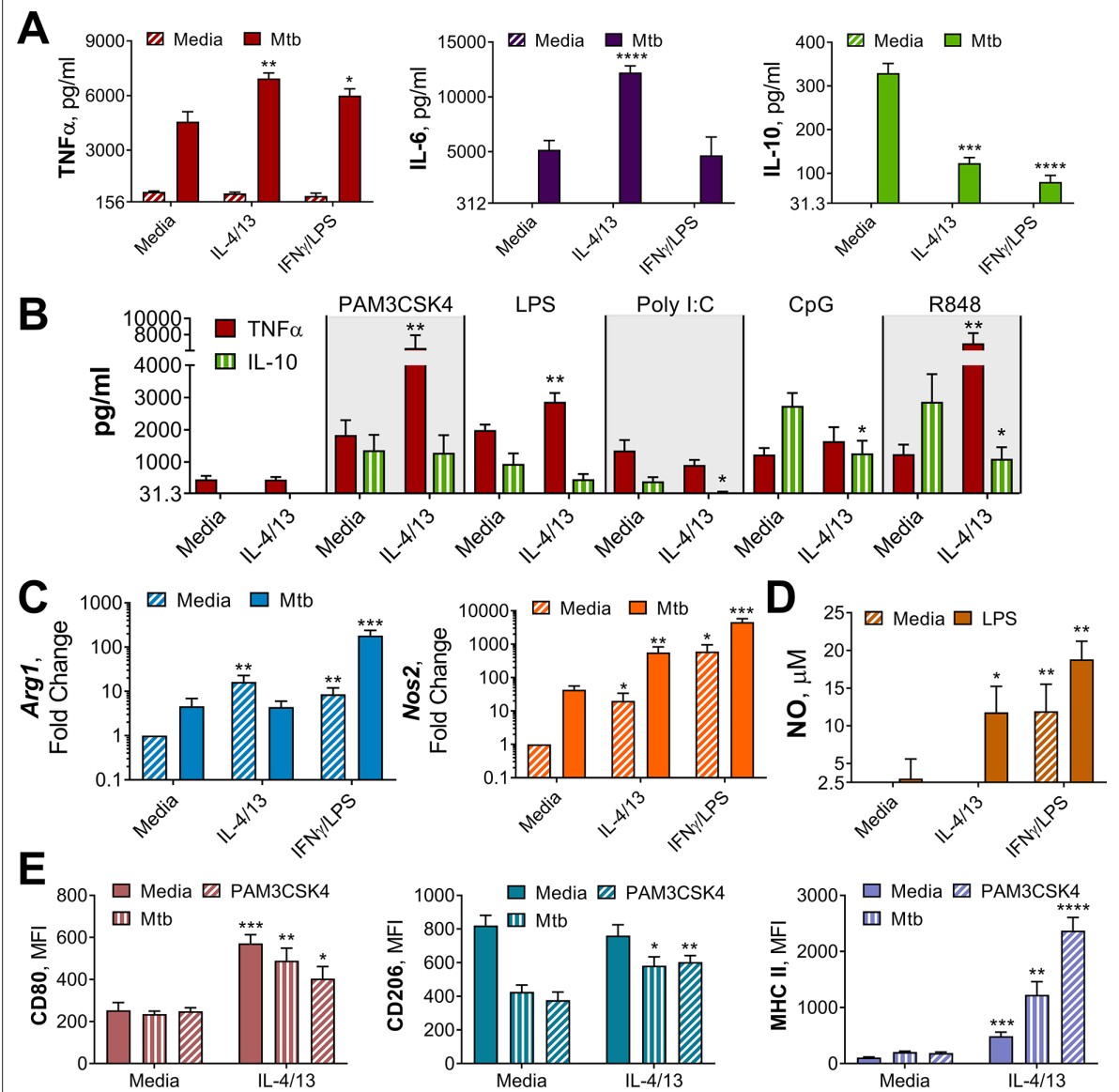

**Figure 2.** Innate training of BMDMs with IL-4 and IL-13 enhances pro-inflammatory responses. (**A–B**) Cytokine secretion following BMDM 24 h incubation with irradiated *M. tuberculosis* (Mtb) (**A**) TLR agonists (**B**) or media on Day 6 as indicated. BMDMs were previously incubated with media, IL-4 with IL-13 or IFNγ with LPS on Day −1 for 24 hr (n=3). (**C**) qPCR of indicated mRNA in BMDMs treated as in (**A**) standardized to BMDMs incubated with media Day −1 and Day 6 (n=4). (**D**) Nitric oxide (NO) secretion from BMDMs treated as in (**B**) (n=3). (**E**) Expression of CD80, CD206 and MHC II (gating strategy Figure S2A) on BMDMs treated as in (**A–B**) (n=3). Mean ± SD are shown and analyzed by student's t-test, compared with media control. * p≤0.05, ** p≤0.01, *** p≤0.001, **** p≤0.0001.

The online version of this article includes the following source code and figure supplement(s) for figure 2:

**Figure supplement 1.** BMDMs either classically activated with IFNγ and LPS or alternatively activated with IL-4 and IL-13.

**Figure supplement 2.** IL-4 and IL-13 innate training enhances inflammatory and bactericidal responses following TLR 1/2 stimulation.

**Figure supplement 3.** Methylation contributes to IL-4 and IL-13 innate training.

**Figure supplement 3—source code 1.** Unedited and labelled western blots.

Next, we addressed whether the shift towards pro-inflammatory responses was induced by either IL-4 or IL-13 alone, or was induced by the two cytokines together, and moreover whether this shift applicable to other stimuli. To address the former, BMDMs were trained with either IL-4, IL-13 or both on Day −1, before incubation with media, Mtb or PAM3CSK4 on Day 6. It was observed that any combination of IL-4 and IL-13 activation enhanced the secretion of TNFα in response to PAM3CSK4,

and that either IL-4 alone or IL-4 with IL-13 significantly reduced the secretion of IL-10 in response to Mtb (*Figure 2—figure supplement 1E*). However, when comparing training with IL-4 alone or training with IL-4 and IL-13, the combination of the two cytokines resulted further attenuation of IL-10 secretion, to a significant extent, therefore the combination of the two cytokines continued to be used. To address whether trained M(4/13) enhanced inflammatory responses was applicable to other stimuli, various Toll-like receptor (TLR) agonists were employed on Day 6 (*Figure 2B*). Incubation of cells with ligands for TLR1/2, TLR4, TLR7/8 or TLR 9 resulted in an increase of TNFα production, reduced IL-10 secretion or both. This demonstrated that prior activation with IL-4 and IL-13 caused a subsequent shift towards a pro-inflammatory response profile in response to a range of pathogen-related agonists.

Having considered cytokine responses, the expression of *Arg1* and *Nos2* and the secretion of NO were next addressed. In mice, the induction of iNOS is important for bactericidal NO production and subsequent killing of *M. tuberculosis* (*Flynn et al., 1993*; *Pasula et al., 2017*). In turn, Arg1 directly impedes the bactericidal function of iNOS by sequestering the amino acid, arginine, which each enzyme uses to make their respective products: ornithine and NO. Subsequently, in the murine model of TB infection, Arg1 has been linked with increased bacterial burden and pathology (*Moreira-Teixeira et al., 2016*).

Without a secondary stimulation on Day 6, M(4/13) and M(IFNγ/LPS) displayed higher levels of *Arg1* and *Nos2* expression, compared with BMDMs incubated in media alone (*Figure 2C*), and M(IFNγ/LPS) exhibited markedly higher expression of *Nos2* compared with M(4/13). Following stimulation with Mtb (*Figure 2C*) or PAM3CSK4 (*Figure 2—figure supplement 2*) M(IFNγ/LPS) exhibited elevated expression of both *Arg1* and *Nos2* compared with untrained M(-), however the trained M(4/13) exhibited equivalent expression of *Arg1*, but with enhanced *Nos2*, demonstrating a further shift towards an M1 profile. An increase in NO production could not be detected following Mtb stimulation; however, following LPS stimulation, both M(4/13) and M(IFNγ/LPS) secreted greater levels of NO compared to M(-) (*Figure 2D*).

Other M1 and M2 markers were analyzed by flow cytometry, following incubation with media, Mtb or PAM3CSK4 on Day 6 (*Figure 2E*). Without secondary stimulation, M(4/13) exhibited an M1-typical profile: heightened expression of CD80 and MHC II, whilst CD206 expression was comparable to M(-). Following secondary stimulation with either Mtb or PAM3CSK4, M(4/13) CD80, CD206, and MHC II expression was significantly greater compared with M(-). Having observed a change in activation markers, cytokine induction and bactericidal capacity in M(4/13) between Days 0 and 6, the next question was whether there was a corresponding shift in glycolytic metabolism.

## BMDMs trained with IL-4 and IL-13 retain OXPHOS metabolism

The increased energy and biosynthetic precursor demand induced by various macrophage activation states are met in distinct ways. To demonstrate this, extracellular flux analysis was carried out, using a mitochondrial stress test. Following basal respiration, oligomycin – an ATP synthase inhibitor – was added to impede electron transport chain (ETC)-driven ATP synthesis, followed by the proton ionophore FCCP (carbonyl cyanide-p-trifluoromethoxyphenylhydrazone), which uncouples the ETC from ATP synthase, allowing maximal respiration, and lastly rotenone and antimycin-A were added, which impede complex I and III of the ETC, respectively. Consistent with previous studies, M(4/13) on Day 1 displayed both an increase of oxygen consumption rate (OCR) – indicative of mitochondrial OXPHOS activity – and extracellular acidification rate (ECAR) – an indirect measurement of lactic acid secretion and thus indicative of glycolysis (*Figure 3A*). This demonstrated how alternative activation is intrinsically linked with enhanced mitochondrial OXPHOS activity, via the tricarboxylic acid (TCA) cycle (*Van den Bossche et al., 2016*; *Wang et al., 2018*). The TCA cycle is in turn driven by glycolysis, glutaminolysis and fatty acid oxidation (*Viola et al., 2019*). By contrast, acutely activated M(IFNγ/LPS) displayed an increase in ECAR and reduced OCR (*Figure 3A*), which was indicative of augmented glycolysis to meet the increased need for ATP, whilst ATP synthesis via OXPHOS is hindered (*Liu et al., 2016*). This glycolytic shift is critical for pro-inflammatory responses induced by classically activated macrophages and moreover results in mitochondrial dysfunction (*Van den Bossche et al., 2016*).

Due to the link between glycolysis and classical activation, it is not surprising that prior work has proposed that a macrophage glycolytic shift is crucial for effective killing and thus overall control of TB infection (*Gleeson et al., 2016*; *Huang et al., 2018*). Furthermore, it has recently been proposed that *M. tuberculosis* impedes this glycolytic shift as an immune evasion strategy (*Hackett et al., 2020*). As

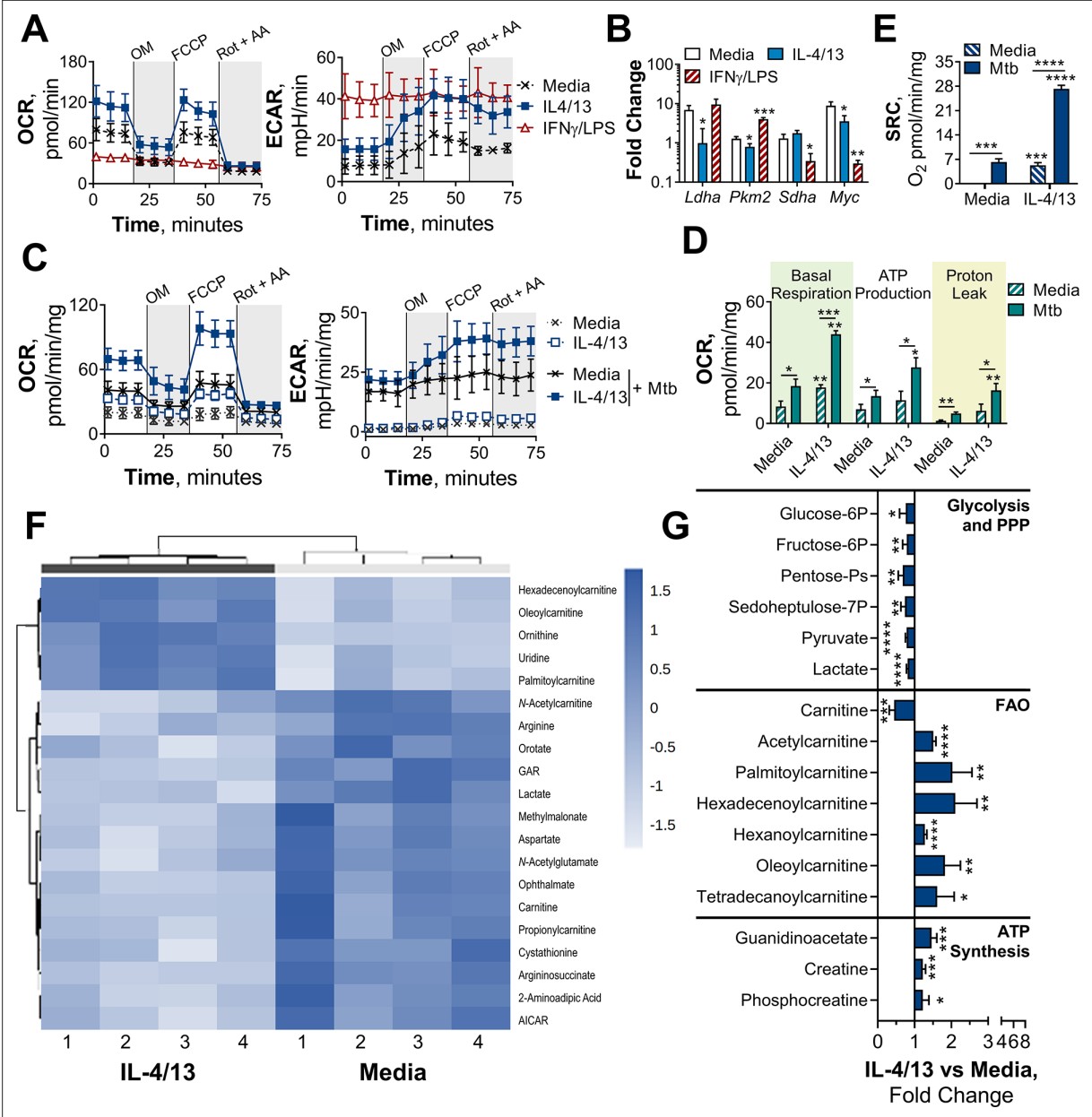

**Figure 3.** BMDMs trained with IL-4 and IL-13 retain M2-typical metabolism upon mycobacterial challenge. (**A**) Extracellular flux analysis of BMDMs, following 24 hr incubation with IL-4 and IL-13, IFNγ and LPS or media (n=3). Mitochondrial stress test inhibitors: oligomycin (OM; blocks ATP synthase), FCCP (uncouples the electron transport chain [ETC] from ATP synthesis), rotenone (Rot; inhibits complex I of the ETC) and antimycin-A (AA; inhibits complex III of the ETC). (**B**) qPCR of indicated mRNA in BMDMs following incubation with media, IL-4 with IL-13 or IFNγ with LPS for 24 hr on Day –1 and stimulated with irradiated *M. tuberculosis* (Mtb) for 6 hr (*Pkm2*) or 24 hr (*Ldha*, *Sdha* and *Myc*), standardized to BMDMs given media on Day –1 and Day 6 (n=3). (**C–E**) Extracellular flux analysis (**C**) basal respiration, ATP production, proton leak (**D**) and spare respiratory capacity (SRC) (**E**) of BMDMs treated as in (**B**) with incubation with media or IL-4 with IL-13 on Day –1 and stimulation with media or Mtb for 24 hr on Day 6 (n=3). (**F–G**) Metabolites from BMDMs treated as in (**B**) incubated with media or IL-4 with IL-13 on Day –1 and stimulation with Mtb for 24 hr on Day 6 (n=4). MetaboAnalyst generated heatmap representing hierarchical clustering of the top 20 most up/down regulated metabolites (**F**). Fold change compared with media control (=1) (**G**). Mean ± SD are shown and analyzed by student's t-test. * p≤0.05, ** p≤0.01, *** p≤0.001, **** p≤0.0001. Abbreviations: AICAR, aminoimidazole carboxamide ribonucleotide; ECAR, extracellular acidification rate; FAO, fatty acid oxidation; GAR, glycinamide ribonucleotide; OCR, oxygen consumption rate; P, phosphate; PPP, pentose phosphate pathway.

The online version of this article includes the following source data and figure supplement(s) for figure 3:

**Source data 1.** MetaboAnalyst R-history.

**Figure supplement 1.** Challenged BMDMs previously trained with IL-4 and IL-13 retain M2-typical metabolism.

we had observed that innate memory responses induced by IL-4 and IL-13 caused a switch towards a pro-inflammatory and bactericidal response profile, akin to a classically activated macrophage, it was pertinent to address whether there was an accompanying glycolytic shift. Previous work by *Van den Bossche et al., 2016* has highlighted that because IL-4 activated human MDMs retain their metabolic versatility, they are able to be 're-polarized' to a classical phenotype. This was demonstrated by activating MDMs for 24 hr with IL-4, before the cells were washed and re-stimulated with IFNγ and LPS for another 24 hr. MDMs previously activated with IL-4 secreted higher concentrations of TNFα, IL-6, and IL-12 compared with MDMs that were naïve prior to IFNγ and LPS stimulation. This adaptive quality to re-polarize is restricted to alternatively activated macrophages, as mitochondrial dysfunction in classically activated macrophages prevents them from re-polarizing to an alternative phenotype (*Van den Bossche et al., 2016*). Subsequently, we next addressed whether the IL-4 trained macrophages were adopting a metabolic profile similar to a classical macrophage, or retaining their more versatile metabolic profile.

As with the BCG infection studies, due to the reduced viability (as observed by reduced cell numbers over time) of the M(IFNγ/LPS) by Day 6 and 7, OCR and ECAR measurements did not reach the detection limit. To compare metabolic phenotypes of the trained macrophages at these later time points, transcription profiles were therefore analyzed. Prior work has established that the transcription factor hypoxia-inducible factor-1 alpha (HIF-1α) aids the glycolytic metabolic shift following classical activation, such as upregulating the enzyme lactate dehydrogenase (LDH) (*Seth et al., 2017*) to increase lactic acid fermentation. Tallying with these studies, trained M(IFNγ/LPS) on Day 7 exhibited elevated expression of LDH (*Ldha*) and pyruvate kinase isozyme M2 (PKM2, *Pkm2*) (*Figure 3—figure supplement 1A*). PKM2 has been shown to play a key role in stabilizing HIF-1α and is thus a crucial determinant for glycolytic metabolism re-wiring (*Palsson-McDermott et al., 2015*). By contrast, on Day 7, M(4/13) displayed a similar level of LDH expression and reduced transcription of PKM2 compared with M(-).

On the other hand, considering OXPHOS machinery at this time point, trained M(IFNγ/LPS) showed reduced expression of the TCA cycle enzyme succinate dehydrogenase (SDH, *Sdha*), whereas this downregulation was not present in M(4/13) (*Figure 3—figure supplement 1A*). Moreover, M(4/13) and M(IFNγ/LPS) respectively displayed increased and decreased expression of the transcription factor myelocytomatosis viral oncogene (c-Myc, *Myc*). IL-4 and IL-13 induce c-Myc expression in alternatively activated macrophages (*Li et al., 2015*; *Luiz et al., 2020*) and contrastingly LPS induced upregulation of HIF-1α occurs in tandem with downregulation of c-Myc (*Liu et al., 2016*). Overall, without secondary stimulation, both trained M(IFNγ/LPS) and M(4/13) retained transcriptional and metabolic profiles consistent with previous reports.

Upon secondary stimulation on Day 6 with either Mtb (*Figure 3B*) or PAM3CSK4 (*Figure 3—figure supplement 1B*), M(IFNγ/LPS) maintained the elevated expression of *Pkm2*, accompanied by reduced transcription of *Sdha* and *Myc. Myc* expression was induced in untrained M(-) and M(4/13) by both secondary stimuli, although Mtb did not enhance *Myc* in M(4/13) to the same extent as M(-). Regarding glycolytic metabolism, M(4/13) displayed relatively reduced *Ldha* upon Mtb incubation (*Figure 3B*) and further maintained a similar level of *Sdha* compared with M(-). These results indicated that trained M(IFNγ/LPS) and M(4/13) maintained their respective metabolic profiles following secondary stimulation with Mtb or PAM3CSK4. As such, these results indicated again that trained M(IFNγ/LPS) and M(IL-4/13) maintained their respective metabolic profiles following secondary stimulation with Mtb or PAM3CSK4.

That trained M(4/13) retain OXPHOS metabolism was furthermore supported by Mtb stimulation increasing both OCR and ECAR in M(-) and M(4/13) (*Figure 3C*), where M(4/13) displayed significantly greater OXPHOS driven basal respiration, ATP production and concomitant proton leakage compared with M(-) (*Figure 3D*). In addition, in response to either Mtb (*Figure 3E*) or PAM3CSK4 (*Figure 3—figure supplement 1C*) there was an increase in spare respiratory capacity (SRC), further signifying that trained M(4/13) retain and even elevate OXPHOS metabolism following secondary stimulation with Mtb.

To further investigate the metabolic profile of trained M(IL-4/13) vs untrained (media control) stimulated with Mtb, the relative change of intracellular metabolite abundance was furthermore assessed by liquid chromatography-mass spectrometry (LC-MS, *Figure 3F–G* and *Figure 3—figure supplement 1D-E*). This semi-targeted analysis revealed that the trained M(4/13) appeared to have increased

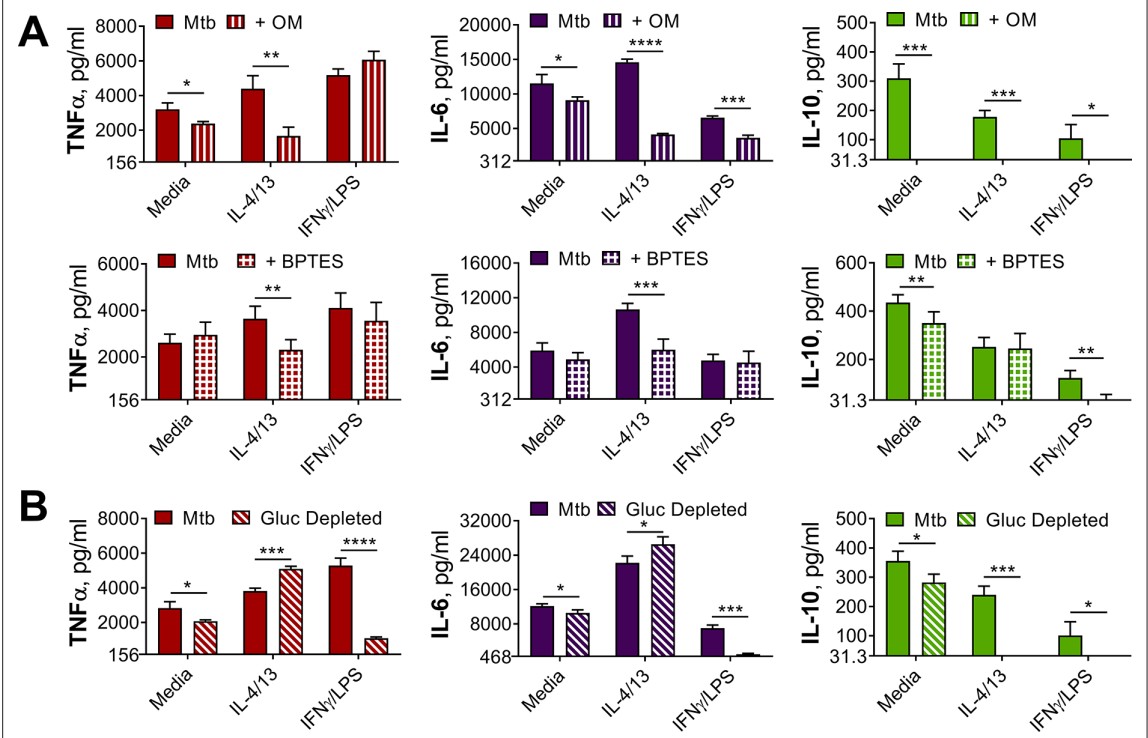

**Figure 4.** OXPHOS drives inflammatory cytokine responses in IL-4/13 trained BMDMs. (**A**) Cytokine secretion from BMDMs following incubation with media, IL-4 with IL-13 or IFNγ with LPS for 24 hr on Day –1 and incubation with irradiated *M. tuberculosis* (Mtb) on Day 6 for 24 hr, with or without pre-incubation (Day 6) of oligomycin (OM) or BPTES (n=4). (**B**) Cytokine secretion from BMDMs treated as in (**A**) with or without glucose depleted conditions between Day –1–7 (n=3). Mean ± SD are shown and analyzed by student's t-test as indicated. * p≤0.05, ** p≤0.01, *** p≤0.001, **** p≤0.0001.

The online version of this article includes the following figure supplement(s) for figure 4:

**Figure supplement 1.** Innate training with IL-4 and IL-13 enhancement of pro-inflammatory responses is not dependent upon glucose metabolism.

use of the urea cycle, as shown by reduced aspartate, arginine and argininosuccinate levels along with enhanced ornithine (*Figure 3—figure supplement 1E*). Furthermore, there were reduced levels of metabolites associated with glycolysis and the pentose phosphate pathway (PPP) – such as lactate and sedoheptulose-7-phosphate – whilst metabolites involved in fatty acid oxidation (FAO) and ATP synthesis regulation were enhanced (*Figure 3G*). In the case of FAO, reduced carnitine with enhanced carnitine-bound fatty acids (acyl carnitines) signified that fatty acids were being ferried by carnitine into the mitochondrion for FAO. FAO is upregulated during murine M2 macrophage activation and results in the production of acetyl-CoA, NADH and FADH$_2$, which are further used to fuel the TCA cycle and downstream OXPHOS (*O'Neill et al., 2016*). Furthermore, the upregulation of creatine and phosphocreatine – as well as creatine biosynthesis precursor guanidinoacetate – intimated enhanced ATP synthesis. Creatine reacts with ATP to form ADP and phosphocreatine, transporting ATP out of the mitochondrion and preventing allosteric inhibition of ATP synthesis. In the trained M(4/13) the amount of phosphocreatine per ATP was enhanced (*Figure 3—figure supplement 1F*), which indicated enhanced mitochondrial ATP synthesis and supported the hypothesis that OXPHOS was enhanced. Overall, these results suggested the trained M(4/13) were upregulating their OXPHOS activity in response to mycobacterial challenge.

Having observed that trained M(IL-4/IL-13) showed a distinct metabolic profile from classically activated macrophages, the roles of glycolysis and OXPHOS in driving cytokine production were next investigated with the use of inhibitors (*Figure 4* and *Figure 4—figure supplement 1*). Incubation with the glycolysis and OXPHOS inhibitor 2-deoxy glucose (2-DG) (*Wang et al., 2018*) prior to stimulation with Mtb (*Figure 4—figure supplement 1A*) significantly reduced TNFα secretion in M(IFNγ/LPS) and M(-). 2-DG also reduced PAM3CSK4 induced TNFα in M(4/13) and M(IFNγ/LPS) (*Figure 4—figure supplement 1B*) and reduced IL-6 secretion in M(4/13) following either Mtb or PAM3CSK4 stimulation. Secretion of anti-inflammatory IL-10 following Mtb stimulation was also reduced by 2-DG

in M(IFNγ/LPS) and M(4/13). This demonstrated that glycolysis, OXPHOS or both were involved in driving cytokine responses to Mtb and PAM3CSK4 in all tested macrophages.

To further examine the role of OXPHOS, the ATP-synthase inhibitor oligomycin (OM) and the glutaminase inhibitor bis-2-(5-phenylacetamido-1,3,4-thiadiazol-2-yl)ethyl sulfide (BPTES) were used prior to secondary stimulation. The role of glutaminolysis in the trained M(IL-4/13) was of interest, as it has previously been highlighted to compensate for inhibition of glycolysis and fuel the TCA cycle during IL-4-induced macrophage activation (*Wang et al., 2018*). In the trained M(4/13), inhibition of either OXPHOS or glutaminolysis significantly reduced TNFα and IL-6 secretion following stimulation with either Mtb (*Figure 4A*) or PAM3CSK4 (*Figure 4—figure supplement 1B*), suggesting that both processes helped drive pro-inflammatory responses. By contrast, in trained M(IFNγ/LPS), although the markedly lower level of IL-6 was further impeded by either inhibitor, the secretion of TNFα was unaffected, supporting M(IFNγ/LPS) use of glycolysis to drive pro-inflammatory responses. OM and BPTES furthermore impeded IL-10 secretion from M(IFNγ/LPS), whereas OM reduced IL-10 secretion in M(4/13) and BPTES did not (*Figure 4A*). In the case of trained M(4/13) this implicated that glutamine metabolism was selectively a driver of pro-inflammatory cytokine responses.

To further cement the differences in glycolytic metabolism between trained M(IFNγ/LPS) and M(4/13), an experiment was carried out where BMDMs were incubated in either glucose depleted media or regular glucose-rich media (both without pyruvate) from Day-1 (*Figure 4B*). Whereas glucose depletion significantly reduced the secretion of TNFα, IL-6 and IL-10 from M(IFNγ/LPS) and M(-), following Mtb stimulation on Day 6, glucose depletion did not impair pro-inflammatory cytokine secretion of M(4/13); instead the secretion of both TNFα and IL-6 was elevated under these conditions, whilst levels of regulatory IL-10 were further reduced, displaying an even greater shift towards pro-inflammatory responses. Furthermore, stimulation with PAM3CSK4 yielded similar results (*Figure 4—figure supplement 1C*), although IL-6 secretion remained unaltered. This highlighted that although trained M(IFNγ/LPS) and M(4/13) had similar response profiles, M(4/13) appeared to retain M2-typical metabolism (*Figure 5*).

## IL-10 negatively regulates innate training

A key consideration regarding alternative macrophage activation and TB infection is the influence of concurrent parasitic disease. Along with IL-4 and IL-13, parasites can induce production of the regulatory cytokine IL-10 (*Gause et al., 2013*; *Roy et al., 2018*), which moreover promotes alternative macrophage activation (*Bystrom et al., 2008*; *Mantovani et al., 2004*). Furthermore, it has been shown that products of the helminth *F. hepatica* can train macrophages to secrete more IL-10 following secondary LPS stimulation (*Quinn et al., 2019*). Given its association with alternative macrophage activation and parasitic infection, we next considered the potential role of IL-10 in IL-4/13 induced macrophage innate memory responses.

BMDMs were activated with IL-4, IL-13 and IL-10 (M(4/13/10)) to compare how this phenotype may have differed from M(4/13). Following secondary stimulation with Mtb, whilst M(4/13) demonstrated elevated TNFα and IL-6 by Day 7 (*Figure 6A*), inflammatory cytokine secretion by M(4/13/10) was reduced on both Day 1 and Day 7. Both M(4/13) and M(4/13/10) secreted lower concentrations of TNFα, IL-6 and IL-10 than M(-) on Day 1 (*Figure 6—figure supplement 1A*). Furthermore, following acute activation, M(4/13/10) had similarly elevated levels of *Arg1* as M(4/13) and a comparably minor induction of *Nos2* (*Figure 6B*). The addition of IL-10 moreover caused even greater augmentation of M2-associated *Chil3* and *Retnla* expression. Accompanying flow cytometry analysis demonstrated that both M(4/13/10) and M(4/13) exhibited reduced expression of CD80, compared with M(-), along with enhanced expression of CD206, where the M(4/13/10) had greater CD206 expression (*Figure 6—figure supplement 1B*). Regarding MHC II, only M(4/13) displayed enhanced expression compared with M(-). The similar upregulation of M2-characteristic markers indicated that M(4/13) and M(4/13/10) were two types of alternatively activated macrophages.

A similar comparison was made at Day 7, where both M(4/13) and M(4/13/10) retained enhanced *Arg1*, *Retnla,* and *Chil3* expression and displayed enhanced *Nos2* expression compared with naïve M(-) (*Figure 6—figure supplement 1C*). Regarding accompanying CD80, CD206, and MHC II expression (*Figure 6—figure supplement 1D*): in all tested conditions, CD206 expression was comparable between M(4/13) and M(4/13/10), whereas CD80 expression was elevated in M(4/13/10). Furthermore, M(4/13) retained an elevated expression of MHC II, which increased two- and fourfold following

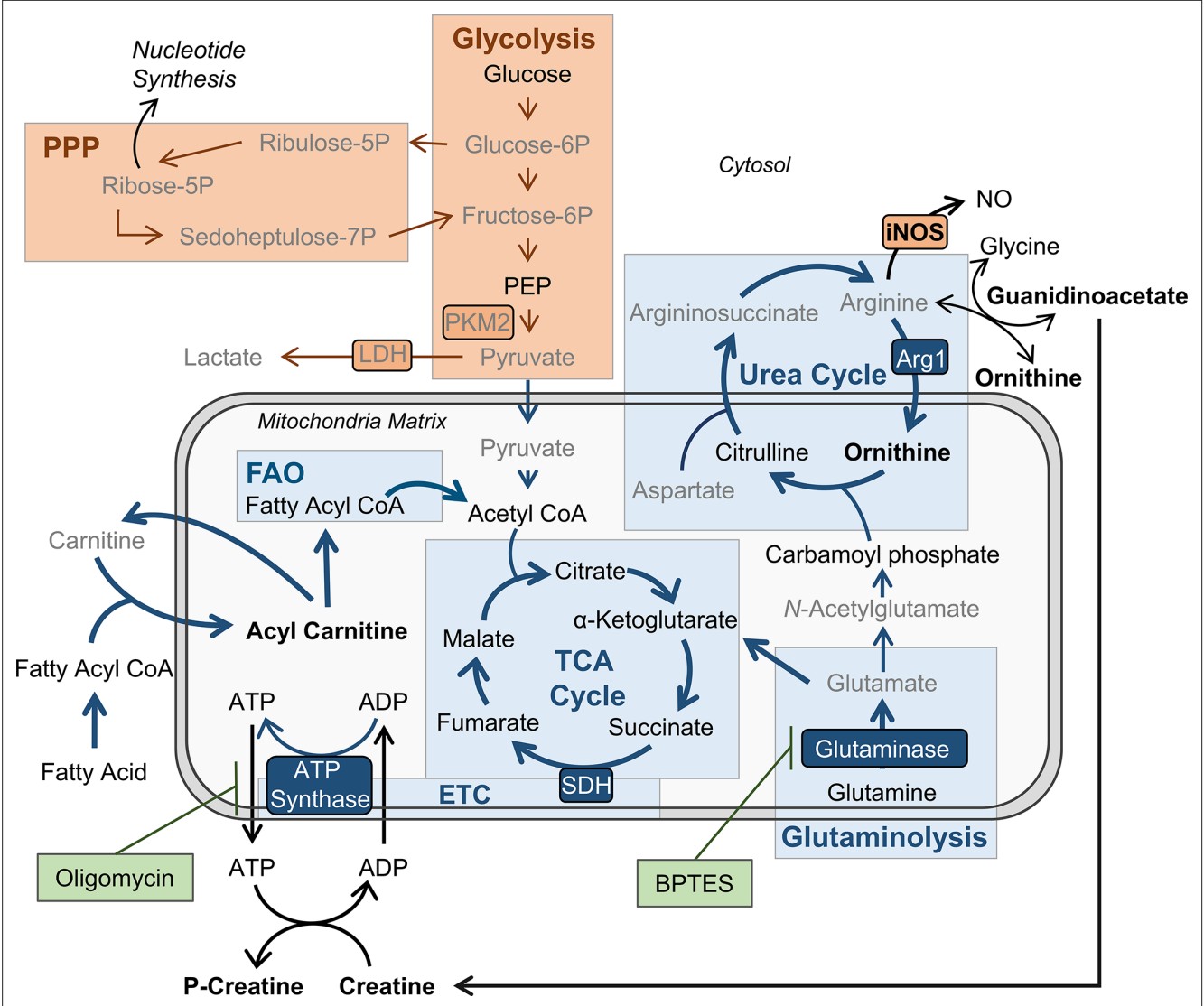

**Figure 5.** BMDMs trained with IL-4 and IL-13 retain M2-typical metabolism: schematic summary of results. Pathways have been simplified. Key: metabolic pathways strongly upregulated by M1/M2 macrophage activation are highlighted by red/blue colored boxes, respectively. Inhibitors are indicated by green boxes. Arrow width represents which pathways are implicated (thicker) or not (narrower) in trained M(4/13) following stimulation with irradiated *M. tuberculosis* (Mtb). Metabolites (measured by LC-MS) or enzymes (measured by qPCR) written in bold or in grey text are enhanced or reduced respectively compared with untrained macrophages. Trained M(4/13) do not employ classical activated macrophage metabolism – aerobic glycolysis and pentose phosphate pathway (PPP) – and instead employ alternative activated macrophage metabolism, characterized by production of ATP through the tricarboxylic acid (TCA) cycle, coupled with the electron transport chain (ETC) via oxidative phosphorylation (OXPHOS), as well as enhanced use of the urea cycle. Glutaminolysis, FAO and ATP synthesis regulation are implicated. This is demonstrated by inhibitor experiments and by changed expression of metabolites. Abbreviations: LDH, lactate dehydrogenase; NO, nitric oxide; P, phosphate; PEP, Phosphoenolpyruvic acid; PKM2, pyruvate kinase M2; SDH, succinate dehydrogenase.

stimulation with Mtb and PAM3CSK4, respectively, whereas this elevation did not occur to the same degree in M(4/13/10). Whilst both M(4/13) and M(4/13/10) moreover displayed similar expression of *Arg1* following Mtb stimulation on Day 7, the addition of IL-10 during activation hindered the upregulation of *Nos2* observed in trained M(4/13), and further enhanced the transcription of *Retnla and Chil3* (*Figure 6—figure supplement 1E*).

Our next question was how the addition of IL-10 during alternative activation would affect mycobacterial killing capacity. The BMDMs were incubated with roughly 30 BCG per cell on Day 0 or Day 6. There were some differences in bacterial uptake (*Figure 6C*), where M(4/13) took up more bacteria per cell on Day 6. Difference in CFU counts after the 3-hr time point (*Figure 6—figure supplement 1A*)

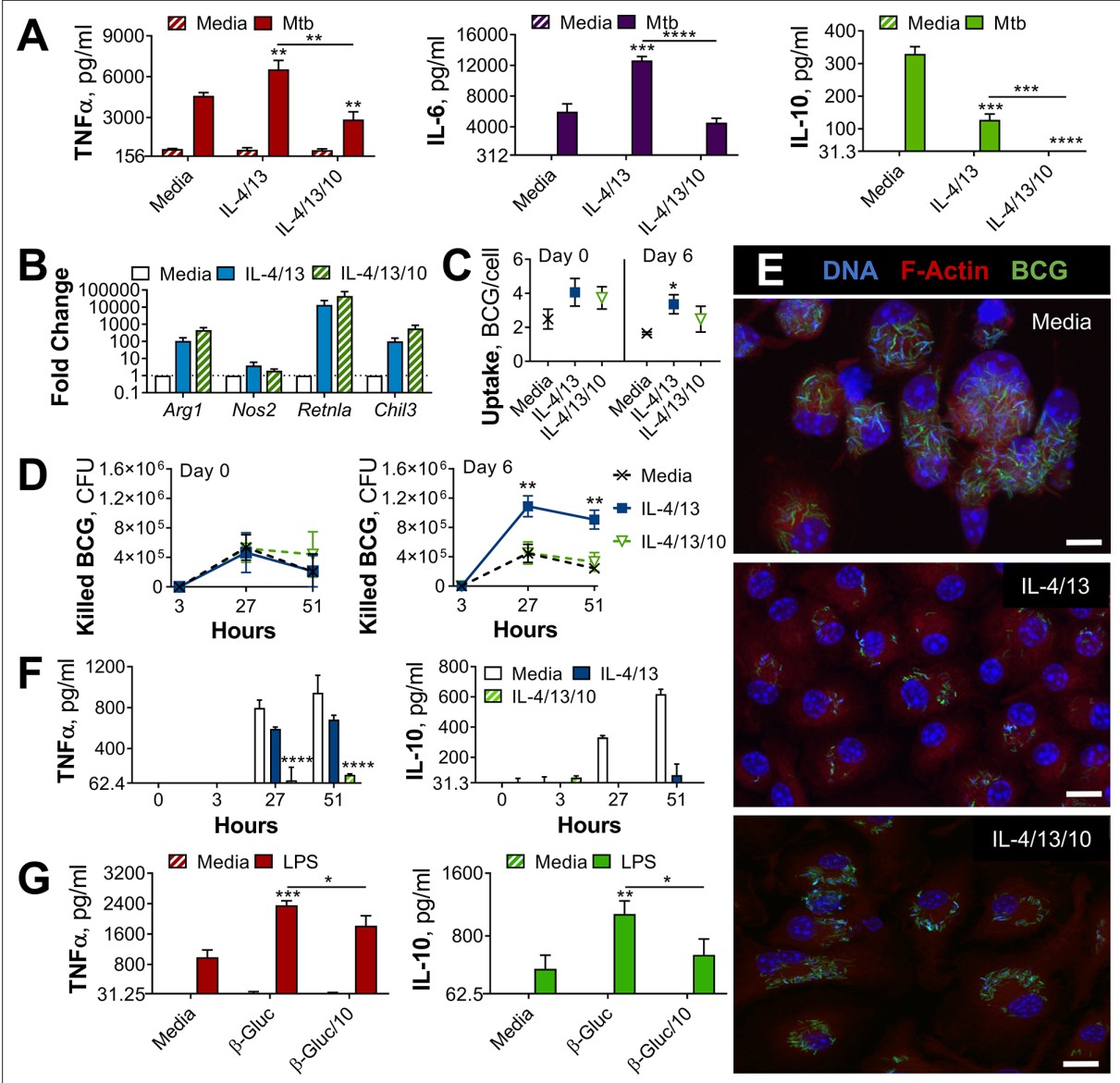

**Figure 6.** IL-10 inhibits bactericidal and pro-inflammatory training induced by IL-4 and IL-13 and alters yeast β-glucan training. (**A**) Cytokine secretion following BMDM incubation with media or IL-4 and IL-13, with or without IL-10, for 24 hr on Day –1 and incubated for 24 hr with media or irradiated *M. tuberculosis* (Mtb) on Day 6. Mean ± SD (n=3) are shown and analyzed by student's t-test, compared to media control or as indicated. (**B**) qPCR of indicated mRNA in BMDMs following 24 hr incubation with media or IL-4 and IL-13, with or without IL-10, standardized to media control. Mean ± SD (n=4) are shown. (**C–F**) BCG Denmark uptake (**C**) killing after h as indicated (**D**) as measured by difference in CFU after 3 hr per 0.5×10⁶ BMDMs on Day 0 or Day 6. Representative images of Hoechst- (blue), modified auramine-O- (green) and phalloidin (red)-stained BMDMs were taken on Day 6, 27 hr after BCG incubation, where scale bars are, from top to bottom, 8- 10- and 10 µm (**E**). Cytokine secretion on Day 6, at h indicated (**F**). BMDMs were previously incubated with media or IL-4 and IL-13, with or without IL-10, for 24 hr on Day –1. (**C–D, F**) representative results (n=2 of ≥3) are shown as mean ± SD (**C, F**) or SEM (**D**) and analyzed by student's t-test compared with media (**C**) or multiple t-tests, with Holm-Sidak correction, comparing with or without IL-10 (**D, F**). (**G**) Cytokine secretion following BMDM incubation with media or β-glucan (β-Gluc), with or without IL-10, for 24 hr on Day –1 and incubated for 6 hr (TNFα) or 24 hr (IL-10) with media or LPS on Day 6. (**A, G**) Mean ± SD (n=3) are shown and analyzed by student's t-test. * p≤0.05, ** p≤0.01, *** p≤0.001, **** p≤0.0001.

The online version of this article includes the following figure supplement(s) for figure 6:

**Figure supplement 1.** Alternative macrophage activation with or without IL-10.

**Figure supplement 2.** CFU counts and cytokine secretion from BCG killing experiments.

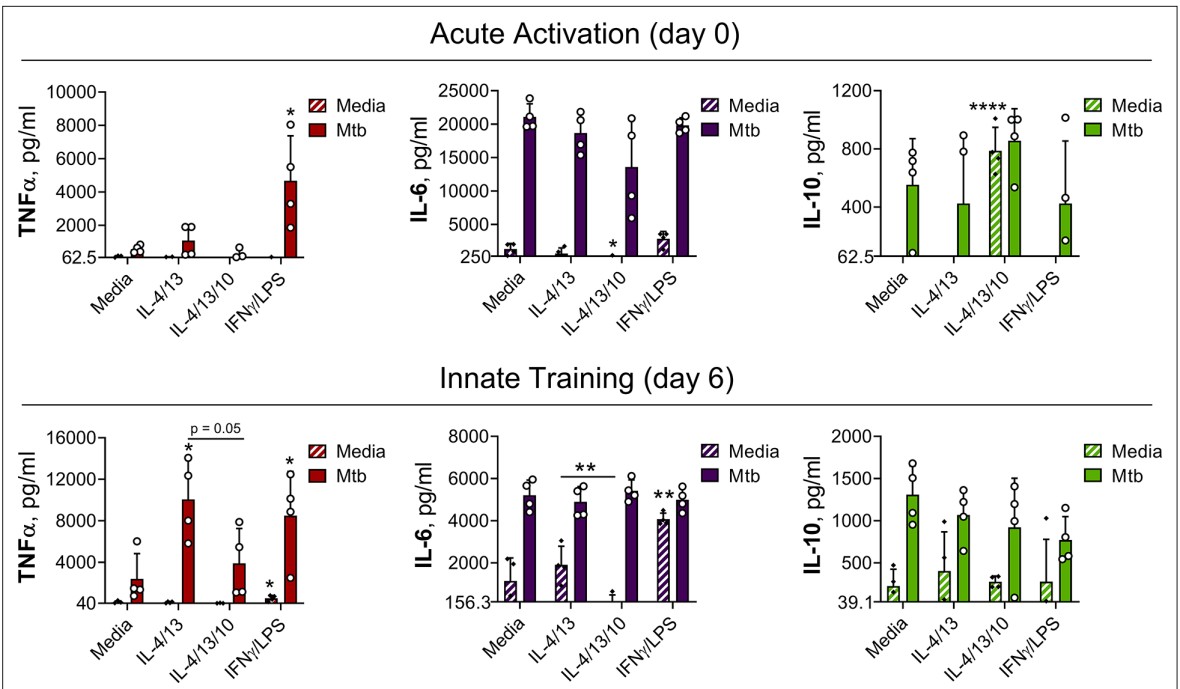

**Figure 7.** IL-4 and IL-13 innate training enhances MoDM secreted TNFα following Mtb stimulation. Cytokine secretion following MoDM 24 hr incubation with irradiated *M. tuberculosis* (Mtb) or media on Day 0 or on Day 6 as indicated. MoDM were previously incubated with media, IL-4 with IL-13 – with or without the addition of IL-10 – or IFNγ with LPS on Day -1 for 24 hr (n = 4, each rep shown by a symbol). Mean ± SD are shown and analyzed by student's t-test, compared with media control or comparing IL-4/IL-13 training with or without IL-10, as indicated. * p ≤ 0.05, ** p ≤ 0.01, **** p ≤ 0.0001.

The online version of this article includes the following figure supplement(s) for figure 7:

**Figure supplement 1.** IL-4 and IL-13 innate training alters MoDM responses following PAM3CSK4 stimulation.

were used to calculate killing of internalized BCG. On Day 0, M(-), M(4/13), and M(4/13/10) displayed comparable killing capacity (*Figure 6D*). Moreover, BCG infection did not induce detectable levels of TNFα or IL-10 (*Figure 6—figure supplement 2B*). On Day 6, M(-) and M(4/13/10) maintained a similar level of BCG killing, whereas M(4/13) displayed enhanced bactericidal capacity. This indicated that IL-10 hindered the enhanced bactericidal response induced by prior IL-4 and IL-13 macrophage activation. The difference in bacterial killing capacity was further supported by confocal microscopy 27 hr after infection, where it was observed that M(-) and M(4/13/10) had markedly more bacteria per cell compared with M(4/13) (*Figure 6E*). Furthermore, whereas M(4/13) showed a shift towards pro-inflammatory cytokine secretion, with IL-10 secretion being reduced whilst TNFα production was largely intact, M(4/13/10) secreted significantly less TNFα than M(4/13), whilst IL-10 secretion was not enhanced (*Figure 6F*). This would indicate that IL-10 regulated the training induced by IL-4 and IL-13, preventing both the enhancement of killing capacity and the accompanying pro-inflammatory cytokine profile.

As IL-10 appeared to impede innate training induced by IL-4 and IL-3, the question of whether IL-10 would impede training induced by other stimuli was investigated. BMDMs were trained with yeast β-glucan – with or without the addition of IL-10 – prior to LPS stimulation a week later (*Figure 6G*). The addition of IL-10 during training reduced the enhanced secretion of both TNFα and IL-10, demonstrating that IL-10 modulates innate training induced by other training stimuli, not only those induced by IL-4 and IL-13.

## IL-4 and IL-13 induced training in human macrophages

Having identified that trained murine M(IL-4/13) show enhanced inflammatory responses following secondary challenge and that this was negatively regulated by IL-10, it was important to test whether these findings were applicable to humans. Human monocyte-derived macrophages (MoDM) were activated with human IL-4 and IL-13, with or without the addition of human IL-10 for 24 hr on Day

–1, and then challenged with irradiated Mtb (*Figure 7*) or PAM3CSK4 (*Figure 7—figure supplement 1*) on Day 0 (acute activation) or Day 6 (innate training). MoDM incubated with media was used as a naïve/untrained control and macrophages activated with human IFNγ and LPS were used to induce classical/M1 activation.

In response to a secondary challenge following acute activation, on Day 0, only MoDMs activated with IFNγ and LPS demonstrated enhanced TNFα secretion (*Figure 7*), whilst MoDMs activated with IL-4 and IL-13 demonstrated similar cytokine levels as the naïve control. At this time point, the addition of IL-10 resulted in significant background secretion of IL-10 and reduced IL-6 secretion following secondary stimulation with PAM3CSK4 (*Figure 7—figure supplement 1*), but otherwise cytokine levels were comparable to the naive control and MoDM activated with IL-4 and IL-13 alone. By contrast, following Mtb stimulation on Day 6, MoDMs trained with either IL-4 and IL-13 or IFNγ and LPS demonstrated significantly enhanced TNFα secretion compared with untrained cells, whilst secretion of IL-6 and IL-10 was unaffected (*Figure 7*). This result was also apparent following secondary challenge with PAM3CSK4 (*Figure 7—figure supplement 1*), although for MoDMs activated with IL-4 and IL-13, this marked increase was not statistically significant. Moreover, the IL-4 and IL-13 induced enhancement of TNFα was impeded by the addition of IL-10.

These results showed that prior activation with IL-4 and IL-13 can specifically enhance inflammatory responses to subsequent challenge, and that this phenomenon is impeded by concurrent IL-10 incubation. This data overall reproduced key results shown previously in the BMDM training model, intimating that these findings may be applicable to human macrophages.

## Discussion

Although 'classical' and 'alternative' activation are used to describe the two extremes of macrophage polarization, the reality is that a broad spectrum of activation states exist, affected by a multitude of signals. Macrophages are key immune cells for determining the outcome of TB disease, and growing evidence indicates that the macrophage population during *M. tuberculosis* infection is highly heterogeneous. Thus, elucidating which subtypes best impede the bacterium is critical for understanding disease control (*Khan et al., 2019*). Alternative macrophage activation is particularly relevant to TB pathology, both because alveolar macrophages – the initial hosts of *M. tuberculosis* upon infection – are biased towards alternative activation (*Huang et al., 2019*), and also considering the influence of concurrent parasitic infection. Given its ultimate relevance in TB pathology, macrophage activation was used as the framework for the work presented herein. The results demonstrate that stimuli of alternative macrophage activation impact innate immune memory, which provides an additional dimension to the noted diversity among macrophage populations.

The current work demonstrates that prior activation with IL-4 and IL-13 enhances anti-mycobacterial responses towards a subsequent challenge, including enhanced inflammatory cytokine secretion (*Figure 2A*) and BCG killing capacity (*Figure 1D*). Regarding how this phenomenon is driven, inhibition of DNA methylation prior to activation with IL-4 and IL-13 impeded the enhanced inflammatory cytokine secretion induced by a secondary challenge, suggesting that this effect is mediated, at least in part, via epigenetic modification (*Figure 2—figure supplement 3A*). However, the role of epigenetic changes – a hallmark for innate training – for the induction of these enhanced responses remains to be addressed in future studies. A role for DNA methylation is in part supported by prior work showing that epigenetic reprogramming is a key component of alternative macrophage activation – crucially de-methylation of H3K27me3 at promoters of key genes such as *Arg1*, *Retnla* and *Chitl3* is required for their expression (*Ishii et al., 2009*; *Liu et al., 2017*). However, whether certain epigenetic changes are sustained, or perhaps altered, between the acute activation stage on Day 0 and the secondary stimulation on Day 6 remains to be investigated. Furthermore, given that epigenetic modification is important for alternative macrophage activation, impeding DNA methylation may have also had a downstream effect upon IL-4 and IL-13 activation, which in turn could have resulted in the reduced inflammatory cytokine secretion to subsequent challenge.

Although further research is required to comprehensively map which mechanism is driving this innate training phenomenon, the current study identifies a macrophage phenotype that is counter-intuitively induced by cytokines previously associated with suboptimal mycobacterial containment. Whilst alternatively activated macrophages have been demonstrated to provide an intracellular environment that is favorable for TB growth (*Kahnert et al., 2006*), resulting in greater bacterial burden

(*Moreira-Teixeira et al., 2016*) in an acute setting, there have been some conflicting data concerning the interplay between parasites and mycobacterial infection. Parasitic infection has been shown to enhance mycobacterial bacterial burden in vivo (*Monin et al., 2015*; *Potian et al., 2011*) and enhance human TB pathology (*Amelio et al., 2017*; *Mabbott, 2018*), but has in some cases also been shown to enhance protection against mycobacterial infection (*Aira et al., 2017*; *O'Shea et al., 2018*). Furthermore, discrepancies have been highlighted specifically regarding macrophage activation. In a model of *M. tuberculosis* macrophage infection in vitro, prior incubation (48 hr) with antigens from *Hymenolepis diminuta*, *Trichuris muris* and *Schistosoma mansoni* resulted in alternative macrophage activation, but only incubation with the *H. diminuta* and *T. muris* antigens resulted in enhanced mycobacterial growth (*Aira et al., 2017*). Moreover, the increased growth was accompanied by enhanced secretion of IL-10. This however was not surprising, given that IL-10 has been demonstrated to prevent phagolysosome maturation in human macrophages (*O'Leary et al., 2011*) and promote TB disease progression in mice (*Beamer et al., 2008*). Herein, an additional mechanism is proposed, where IL-10 reduces IL-4 and IL-13 induced macrophage training and subsequent control of mycobacterial challenge.

BMDM activation with IL-4 and IL-13, with or without the addition of IL-10, resulted in similar activation states: comparable expression of *Arg1*, *Retnla* and *Chil3* on Day 0 (*Figure 6B*) and 7 (*Figure 6—figure supplement 1C***7**), reduced expression of CD80 and upregulation of CD206 on Day 0 (*Figure 6—figure supplement 1B*), as well as reduced secretion of cytokines TNFα, IL-6 and IL-10 following stimulation with Mtb on Day 0 (*Figure 6—figure supplement 1A***A**). Whilst this hyporesponsive profile was maintained in the M(4/13/10), the trained M(4/13) on the other hand demonstrated enhanced pro-inflammatory and bactericidal mechanisms in response to mycobacterial challenge a week after initial activation (*Figure 6*). This intimated that IL-10 was a negative regulator of innate training induced by IL-4 and IL-13 and indicates that distinct cytokine-mediated modes of alternative macrophage activation differ regarding their innate memory programming. The ability of IL-10 to modulate innate training was herein further demonstrated by investigating innate memory induced by β-glucan (*Figure 6G*), where the addition of IL-10 during the training stage resulted in attenuated cytokine secretion following secondary LPS stimulation. This discovery has thereby revealed an exciting avenue for future research in this field.

In response to mycobacterial challenge the IL-4 and IL-13 trained cells adopted a phenotype similar to classically activated BMDMs: a skewing towards pro-inflammatory cytokine secretion with concomitant increased expression of *Nos2* and NO production (*Figure 2*), as well as enhanced mycobacterial killing capacity (*Figure 1*). Classical macrophage activation is intrinsically linked with a glycolytic shift in metabolism; it is thus not surprising that prior work has identified an enhancement of glycolysis as critical for efficient *M. tuberculosis* killing (*Gleeson et al., 2016*; *Huang et al., 2018*), and that impeding glycolysis attenuates the ability of macrophages to control TB infection (*Hackett et al., 2020*). Moreover, upon *M. tuberculosis* infection, macrophages appear to undergo a biphasic metabolic profile, where they switch from an initial increase in glycolytic metabolism to an enhancement of the TCA cycle and OXPHOS, a switch which allows mycobacterial survival and disease progression (*Shi et al., 2019*). As such, it was expected that the trained M(4/13) would shift towards glycolytic metabolism. However, this was not the case, as Mtb stimulation resulted in enhanced OCR and SRC, indicating the use and upregulation of OXPHOS (*Figure 3*). Furthermore, LC-MS analysis of metabolites indicated increased use of FAO, as well as enhanced creatine, further supporting the continued use of M2-typical metabolism. This was moreover confirmed with the use of the ATP synthase inhibitor oligomycin, where it was demonstrated that inhibition of OXPHOS reduced trained M(4/13) cytokine secretion; by contrast, TNFα upregulation in M(IFNγ/LPS) was unaffected. Furthermore, when trained M(4/13) were incubated in a glucose depleted environment during the week preceding secondary stimulation, the pro-inflammatory shift was enhanced as reflected in increased TNFα and IL-6 secretion combined with reduced IL-10 production (*Figure 4B*). This was in stark contrast to the effect of glucose depletion on untrained M(-) and trained M(IFNγ/LPS), where the secretion of all measured cytokines was attenuated. This contrast between M(4/13) and M(IFNγ/LPS) cemented their differences regarding metabolic dependency on glucose. It should be noted that prior work has identified differences in macrophage metabolic profiles in response to infection with live *M. tuberculosis* compared with the killed bacterium or infection with attenuated BCG (*Cumming et al., 2018*). However, the consensus is that classic macrophage activation, and an accompanying shift to glycolytic metabolism, is paramount for effective mycobacterial killing and propagation of inflammatory responses. It is surprising

therefore that these results summarily intimated that trained M(4/13) do not rely on glycolysis to fuel anti-mycobacterial responses.

The addition of a glycolysis inhibitor reduced cytokine secretion, from M(-), M(IFNγ/LPS), as well as M(4/13) following secondary stimulation (Figure 3—figure supplement 2A-**B**). That inhibition of glycolysis impeded pro-inflammatory cytokine secretion, but glucose depletion did not, indicated that glycolysis in trained M(4/13) was fueling the TCA cycle and downstream OXPHOS, and upon glucose depletion, other pathways, such as glutaminolysis, were compensating for this loss. Glutaminolysis has been shown to be upregulated following macrophage activation with IL-4 and has been demonstrated to compensate for glycolysis inhibition in driving the TCA cycle and OXPHOS (*Wang et al., 2018*). In the present study, it was moreover observed that the use of a glutaminolysis inhibitor reduced TNFα and IL-6 secretion following secondary stimulation in trained M(4/13) specifically, and not in untrained M(-) or trained M(IFNγ/LPS) (*Figure 4A*). This work suggests that OXPHOS can be important for pro-inflammatory responses, an observation that is supported by previous work demonstrating that mitochondrial stress is a mechanism of macrophage tolerance (*Timblin et al., 2021*) and that the ETC is important for nod-like receptor family pyrin domain containing 3 (NLRP3) inflammasome activity (*Billingham et al., 2022*). In the latter study, however, LPS induced TNFα secretion was not impeded by inhibition of ETC complexes I and II, demonstrating once more that the metabolic phenotype of the trained M(4/13), explored in the current study, is highly unexpected and warrants further study.

Regarding classical macrophage activation, in the current study concurrent LPS and IFNγ stimulation did not appear to lead to tolerance, as secondary stimulation with Mtb on Day 0 (*Figure 2—figure supplement 1D*) or Day 6 (*Figure 2A*) resulted in enhanced TNFα and dampened IL-10 secretion. This is somewhat contrary to prior work demonstrating how LPS induces tolerance in macrophages (*Domínguez-Andrés et al., 2019*; *Timblin et al., 2021*). This discrepancy may be due to the relatively low concentration of LPS employed in the present study, and/or the concurrent stimulation with IFNγ. However, although tolerance was not induced, for the classically activated BMDMs and MoDMs there was a reduction in viability over time; as a result there were too few cells to reach the detection limit to measure extracellular flux in the trained M(IFNγ/LPS) BMDM. It could be that the metabolic switch to aerobic glycolysis results in a macrophage phenotype that cannot be sustained over such a long period in culture. Overall, this presumed loss of viability greatly limited the number of experiments where trained M(IL-4/13) could be directly compared with trained M(IFNγ/LPS). Although further investigation of long-term activation states of M(IFNγ/LPS) are warranted, the accumulated results for M(IFNγ/LPS) in the current study demonstrate the distinctive metabolic phenotype driving the inflammatory responses in the trained M(IL-4/13).

The metabolic profile in trained M(4/13), as summarized in *Figure 5*, is not only distinct from classically activated macrophages, but also from that seen with other training stimuli, such as β-glucan. Innate training is being tested as a means to bolster innate immunity against *M. tuberculosis* (*Khader et al., 2019*) and it was recently reported that training with β-glucan was protective against subsequent *M tuberculosis* infection, as seen by enhanced human monocyte pro-inflammatory cytokine secretion and increased mouse survival in vivo (*Moorlag et al., 2020*). Similar to classical macrophage activation, β-glucan training results in a glycolytic shift, as demonstrated in human monocytes (*Cheng et al., 2014*). As such, the phenotype of the trained M(4/13) is distinct from other macrophage phenotypes previously demonstrated to protect against TB, and thus offers an additional avenue for future research regarding strategies for combatting this disease.

The majority of the presented experiments were carried out in a murine in vitro system, using live BCG or irradiated *M. tuberculosis* H37Rv. Therefore, future work should address the translation of our findings to TB in vivo models, as well as in the context of human TB disease pathology. Regarding metabolic profiles, prior work has identified differences in human macrophage metabolic phenotypes in response to infection with live virulent *M. tuberculosis* compared with the killed bacterium or infection with BCG; BCG induced greater glycolytic metabolism, whilst infection with virulent *M. tuberculosis* H37Rv resulted in a quiescent phenotype where both extracellular acidification and oxygen consumption were dampened with increasing MOI (*Cumming et al., 2018*). Although preliminary work herein has demonstrated that human monocyte derived macrophages trained with IL-4 and IL-13 show a markedly enhanced secretion of TNFα in response to secondary stimulation with either irradiated Mtb (*Figure 7*) or PAM3CSK4 (*Figure 7—figure supplement 1*), the metabolic profile driving

these responses remains to be investigated, as well as how this macrophage activation state would impact on the outcome of infection with virulent *M. tuberculosis*.

*Van den Bossche et al., 2016* highlighted a key adaptive distinction between classical and alternative macrophage activation: that alternatively activated macrophages can be 're-polarized' due to their metabolic versatility, whereas the mitochondrial dysfunction which occurs during classical macrophage activation prevents such reprogramming. The current study identifies an additional adaptive capacity where activation of macrophages with IL-4 and IL-13 programs BMDMs to better respond to a subsequent mycobacterial challenge, whilst crucially retaining their metabolic diversity. In conclusion, our work presents a new insight into how innate training with IL-4 and IL-13 can enhance macrophage pro-inflammatory responses and mycobacterial killing. This unexpected finding shows how macrophage plasticity belies the usual M1-M2 dichotomy and provides a new framework to explore the impact of comorbidities, such as parasitic infections, on TB disease burden.

# Materials and methods

Key resources table

| Reagent type (species) or resource | Designation | Source or reference | Identifiers | Additional information |
|---|---|---|---|---|
| Strain, strain background (*Escherichia coli*, serotype R515) | Lipopolysaccharide, LPS | Enzo | Cat# ALX-581–007 L002 | |
| Strain, strain background (*H. sapiens*) | Primary cell isolation | Buffy packs from Irish Blood Transfusion Service | N/A | |
| Strain, strain background (*M. musculus*, C57BL/6JOlaHsd) | Primary cell isolation | In-house colonies | C57BL/6JOlaHsd | Both sexes employed |
| Strain, strain background (*Mycobacterium bovis*) | Bacille Calmette-Guérin (BCG) Denmark 1331 | Gift to Prof. Gordon from Prof. Behr, McGill University, Canada | N/A | |
| Strain, strain background (*Mycobacterium tuberculosis*, strain H37Rv) | Irradiated whole cells of *M. tuberculosis* | BEI resources | NR-49098 | Non-viable bacteria |
| Cell line (*M. musculus*) | L929 | gift of Prof. Muñoz-Wolf, Trinity College, Dublin gift of Prof. Sheedy, Trinity College, Dublin | N/A | Cell lines maintained in E. Lavelle and F. Sheedy labs. |
| Antibody | Anti-CD11b-APC-eFluor 780 (Rat monoclonal) | Thermo Fisher Scientific | Cat# 47-0112-82, Clone M1/70 | FACS (0.1 µl per test) |
| Antibody | Anti-CD206-PE (Rat monoclonal) | BioLegend | Cat# 141706, Clone C068C2 | FACS (0.4 µl per test) |
| Antibody | Anti-CD80-FITC (Hamster monoclonal) | BD Biosciences | Cat# 561954, Clone 16–10 A1 | FACS (0.15 µl per test) |
| Antibody | Anti-F4/80-PerCP-Cy5.5 (Rat monoclonal) | Thermo Fisher Scientific | Cat# 45-4801-82, Clone BM8 | FACS (0.25 µl per test) |
| Antibody | Anti-Histone H3 (Mouse monoclonal) | Active Motif | Cat# 39763 | WB (1/3,000) |
| Antibody | Anti-H3K4me3 (Rabbit polyclonal) | Abcam | Cat# ab8580 | WB (1/1000) |
| Antibody | Anti- H3K9me2 (Mouse monoclonal) | Abcam | Cat# ab1220 | WB (1/1000) |
| Antibody | Anti-H3K27me3 (Mouse monoclonal) | Active Motif | Cat# 61017 | WB (1/1000) |

*Continued on next page*

*Continued*

| Reagent type (species) or resource | Designation | Source or reference | Identifiers | Additional information |
|---|---|---|---|---|
| Antibody | Anti-MHC class II-eFlour 450 (Rat monoclonal) | Thermo Fisher Scientific | Cat# 48-5321-82, Clone M5/114.15.2 | FACS (0.2 µl per test) |
| Antibody | Anti-mouse IgG (Goat monoclonal) | LI-COR | Cat# 925–32210 | WB (1/5000 dilution) |
| Antibody | Anti-Rabbit IgG (Goat monoclonal) | LI-COR | Cat# 926–32211 | WB (1/2500) |
| Antibody | Fc block: Anti-CD16/CD32 (Rat monoclonal) | BD Biosciences | Cat# 553142, Clone 2.4G2 | FACS (0.5 µl per test) |
| Sequence-based reagent Actb _F | | Primer BLAST | Invitrogen Custom DNA Oligos | GCTTCTTTGCAGCTCCTTCGT |
| Sequence-based reagent Actb _R | | Primer BLAST | Invitrogen Custom DNA Oligos | CGTCATCCATGGCGAACTG |
| Sequence-based reagent Arg1 _F | | Primer BLAST | Invitrogen Custom DNA Oligos | TACAAGACAGGGCTCCTTTCAG |
| Sequence-based reagent Arg1 _R | | Primer BLAST | Invitrogen Custom DNA Oligos | TGAGTTCCGAAGCAAGCCAA |
| Sequence-based reagent Chitl3 _F | | Primer BLAST | Invitrogen Custom DNA Oligos | AAGCTCTCCAGAAGCAATCC |
| Sequence-based reagent Chitl3 _R | | Primer BLAST | Invitrogen Custom DNA Oligos | AGAAGAATTGCCAGACCTGTGA |
| Sequence-based reagent Ldha _F | | Primer BLAST | Eurofins genomics (MWG) | GAGACTTGGCTGAGAGCATAA |
| Sequence-based reagent Ldha _R | | Primer BLAST | Eurofins genomics (MWG) | GATACATGGGACACTGAGGAA |
| Sequence-based reagent Myc _F | | Primer BLAST | Invitrogen Custom DNA Oligos | CAGCGACTCTGAAGAAGAGCA |
| Sequence-based reagent Myc _R | | Primer BLAST | Invitrogen Custom DNA Oligos | GACCTCTTGGCAGGGGTTTG |
| Sequence-based reagent Nos2 _F | | Primer BLAST | Invitrogen Custom DNA Oligos | TCCTGGACATTACGACCCCT |
| Sequence-based reagent Nos2 _R | | Primer BLAST | Invitrogen Custom DNA Oligos | CTCTGAGGGCTGACACAAGG |
| Sequence-based reagent Pkm2 _F | | Primer BLAST | Eurofins genomics (MWG) | TGTCTGGAGAAACAGCCAAG |
| Sequence-based reagent Pkm2_R | | Primer BLAST | Eurofins genomics (MWG) | CGAATAGCTGCAAGTGGTAGA |
| Sequence-based reagent Retnla _F | | Primer BLAST | Invitrogen Custom DNA Oligos | CAGCTGATGGTCCCAGTGAAT |
| Sequence-based reagent Retnla _R | | Primer BLAST | Invitrogen Custom DNA Oligos | AGTGGAGGGATAGTTAGCTGG |
| Sequence-based reagent Sdha _F | | Primer BLAST | Eurofins genomics (MWG) | GGAACACTCCAAAAACAGACC |
| Sequence-based reagent Sdha _R | | Primer BLAST | Eurofins genomics (MWG) | CCACCACTGGGTATTGAGTAGAA |
| Sequence-based reagent Tbp _F | | Primer BLAST | Invitrogen Custom DNA Oligos | CAGGAGCCAAGAGTGAAGAACA |
| Sequence-based reagent Tbp _R | | Primer BLAST | Invitrogen Custom DNA Oligos | AAGAACTTAGCTGGGAAGCCC |
| Peptide, recombinant protein | Heat-Shocked Bovine Serum Albumin (BSA) | Thermo Fisher Scientific | Cat# 12881630 | |
| Peptide, recombinant protein | M-MLV reverse transcriptase | Promega | Cat# M3683 | |

*Continued on next page*

*Continued*

| Reagent type (species) or resource | Designation | Source or reference | Identifiers | Additional information |
|---|---|---|---|---|
| Peptide, recombinant protein | Recombinant human IFNγ | Peprotech | Cat# 300–02 | |
| Peptide, recombinant protein | Recombinant human IL-10 | Peprotech | Cat# 200–10 | |
| Peptide, recombinant protein | Recombinant human IL-13 | Peprotech | Cat# 200–13 | |
| Peptide, recombinant protein | Recombinant human IL-4 | Peprotech | Cat# 200–04 | |
| Peptide, recombinant protein | Recombinant human M-CSF | Prospec Protein Specialists | Cat# CYT-308 | |
| Peptide, recombinant protein | Recombinant murine IFNγ | Peprotech | Cat# 315–05 | |
| Peptide, recombinant protein | Recombinant murine IL-10 | Peprotech | Cat# 210–10 | |
| Peptide, recombinant protein | Recombinant murine IL-13 | Peprotech | Cat# 210–13 | |
| Peptide, recombinant protein | Recombinant murine IL-4 | Peprotech | Cat# 214–14 | |
| Commercial assay or kit | BCA Protein Assay Kit (Pierce) | Thermo Fisher Scientific | Cat# 23225 | |
| Commercial assay or kit | Griess Reagent System kit | Promega | Cat# G2930 | |
| Commercial assay or kit | High Pure RNA Isolation Kit | Roche | Cat# 11828665001 | |
| Commercial assay or kit | Human IL-10 ELISA Kit uncoated | Invitrogen | Cat# 88710688 | |
| Commercial assay or kit | Human IL-6 ELISA Kit uncoated | Invitrogen | Cat# 88706688 | |
| Commercial assay or kit | Human TNFα ELISA Kit uncoated | Invitrogen | Cat# 88734688 | |
| Commercial assay or kit | Mouse IL-10 ELISA MAX | BioLegend | Cat# 431411 | |
| Commercial assay or kit | Mouse IL-6 ELISA MAX | BioLegend | Cat# 431301 | |
| Commercial assay or kit | Mouse TNFα DuoSet ELISA | R&D Systems | Cat# DY410 | |
| Chemical compound, drug | 2-deoxyglucose, 2-DG | Sigma Aldrich | Cat# D8375 | |
| Chemical compound, drug | Antimycin-A | Sigma Aldrich | Cat# A8674 | |
| Chemical compound, drug | BPTES | Sigma Aldrich | Cat# SML0601-5mg | |
| Chemical compound, drug | CpG | InvivoGen | Cat# ODN M362 | |
| Chemical compound, drug | FCCP | Sigma Aldrich | Cat# C2920 | |
| Chemical compound, drug | Glucose | Sigma Aldrich | Cat# G8270 | |
| Chemical compound, drug | L-Glutamine | Gibco | Cat# 25030–024 | |
| Chemical compound, drug | Glycerol | Sigma Aldrich | Cat# G2025 | |

*Continued on next page*

*Continued*

| Reagent type (species) or resource | Designation | Source or reference | Identifiers | Additional information |
|---|---|---|---|---|
| Chemical compound, drug | MTA | Sigma Aldrich | Cat# D5011-25MG | |
| Chemical compound, drug | Oligomycin | Sigma Aldrich | Cat# 75351 | |
| Chemical compound, drug | PAM3CSK4 | InvivoGen | Cat# tlrl-pms | |
| Chemical compound, drug | Poly I:C | InvivoGen | Cat# tlrl-pic | |
| Chemical compound, drug | Pyruvate | Sigma Aldrich | Cat# P5280 | |
| Chemical compound, drug | Resiquimod/R848 | InvivoGen | Cat# tlrl-r848 | |
| Chemical compound, drug | Rotenone | Sigma Aldrich | Cat# R8875 | |
| Chemical compound, drug | Sodium Chloride | Sigma Aldrich | Cat# S9888 | |
| Chemical compound, drug | Valine-d8 | CK isotopes | Cat# DLM-488 | |
| Chemical compound, drug | WGP Dispersible | InvivoGen | Cat# tlrl-wgp | |
| Software, algorithm | FlowJo 7 | FlowJo LLC, Franklin Lakes, New Jersey | https://www.flowjo.com/solutions/flowjo | |
| Software, algorithm | ImageJ | National Institutes of Health and the Laboratory for Optical and Computational Instrumentation | https://imagej.nih.gov/ij/ | |
| Software, algorithm | MetaboAnalyst 5.0 | Xia Lab @ McGill | https://www.metaboanalyst.ca/ | |
| Software, algorithm | Microsoft Office Excel | Microsoft, Redmond, Washington | https://products.office.com/en-au/excel | |
| Software, algorithm | Prism 8.2 | GraphPad Software, San Diego, California | https://www.graphpad.com/scientific-software/prism/ | |
| Software, algorithm | Tracefinder 5.0 | Thermo Fisher Scientific, Waltham, Massachusetts | https://www.thermofisher.com/ie/en/home/industrial/mass-spectrometry/liquid-chromatography-mass-spectrometry-lc-ms/lc-ms-software/lc-ms-data-acquisition-software/tracefinder-software.html | |
| Other | 4% PFA in PBS | Santa Cruz Biotechnology | Cat# NC0238527 | Fixing buffer for flow cytometry and confocal microscopy |
| Other | Acetonitrile | Thermo Fisher Scientific | Cat# 10001334 | Extraction buffer for Metabolomics |
| Other | DMEM (high glucose) | Sigma Aldrich | Cat# D5671 | Cell culture media (BMDMs) |
| Other | DMEM (no glucose) | Gibco | Cat# 11966025 | Cell culture media (BMDMs) |
| Other | dNTP Mix | Meridian Bioscience | Cat# BIO-39028 | Nucleotides for cDNA synthesis |

*Continued on next page*

*Continued*

| Reagent type (species) or resource | Designation | Source or reference | Identifiers | Additional information |
|---|---|---|---|---|
| Other | FBS | Biosera | Batch# 015BS551 | Serum for cell culture media |
| Other | Fixable Viability Stain 510 | Invitrogen | Cat# 564406 | Viability stain used for FACS |
| Other | Hoechst 33342 | Thermo Fisher Scientific | Cat# 10150888 | DNA dye used for confocal microscopy |
| Other | KAPA SYBR FAST Rox low qPCR Kit Master Mix | Sigma Aldrich | Cat# KK4622 | Nucleic acid stain for qPCR |
| Other | Lymphoprep | Stemcell Technologies | Cat# 07851 | Density gradient medium for isolating PBMCs |
| Other | Methanol | Thermo Fisher Scientific | Cat# 10284580 | Extraction buffer for Metabolomics |
| Other | Middlebrook 7H11 powder | Sigma Aldrich | Cat# M0428 | For making Mycobacterial culture media (agar plates) |
| Other | Middlebrook 7H9 powder | Sigma Aldrich | Cat# M0178 | For making Mycobacterial culture media |
| Other | Modified Auramine-O stain and quencher | Scientific Device Laboratory | Cat# 345–04 L | Mycobacterial stain for confocal micrsocopy |
| Other | PBS, sterile | Gibco | Cat# 14190094 | Cell wash buffer |
| Other | Penicillin-Streptomycin | Gibco | Cat# 15-070-063 | Antibiotics for cell culture |
| Other | Phalloidin-Alexa Fluor 647 | Invitrogen | Cat# A22287 | Stain actin for confocal microscopy |
| Other | Radio-Immunoprecipitation Assay (RIPA) buffer | Sigma Aldrich | Cat# R0278-50ML | Cell lysis buffer |
| Other | Random Hexamer Primer Mix | Meridian Bioscience | Cat# BIO-38028 | Primers for cDNA synthesis |
| Other | Reverse Transcriptase Buffer | Promega | Cat# A3561 | Buffer for cDNA synthesis |
| Other | RNAseOUT | Invitrogen | Cat# 10777019 | RNAse inhibitor for cDNA synthesis |
| Other | RPMI 1640 Glutamax | Gibco | Cat# 21875034 | Cell culture media (MoDMs) |
| Other | Seahorse Calibration Fluid pH 7.4 | Agilent | Part# 100840–000 | Extracellular flux calibration fluid |
| Other | Seahorse XF DMEM Medium | Agilent | Cat# 103575–100 | Extracellular flux culture media |
| Other | Tween20 | Sigma Aldrich | Cat# P1379-1L | Detergent for cell lysis and ELISA wash buffer |
| Other | Vectashield mounting media | VWR | Cat# 101098–042 | Mounting media for confocal microscopy |
| Other | Water, sterile | Baxter | Cat# UKF7114 | Solvent |

## Experimental model and subject details

### Animals

Mice used for primary cell isolation were eight to 16-week-old wild-type C57BL/6 mice that were bred in the Trinity Biomedical Sciences Institute Bioresources Unit. Animals were maintained according to the regulations of the Health Products Regulatory Authority (HPRA). Animal studies were approved by the TCD Animal Research Ethics Committee (Ethical Approval Number 091210) and were performed under the appropriate license (AE191364/P079).

### Cell isolation and culture

Bone-marrow-derived macrophages (BMDMs) were generated as described previously by our group (*Lebre et al., 2018*). Briefly, bone marrow cells were extracted from the leg bones and were cultured

in high glucose DMEM, supplemented with 8% v/v fetal bovine serum (FBS), 2 mM L-glutamine, 50 U ml$^{-1}$ penicillin, 50 µg ml$^{-1}$ streptomycin (hereafter referred to as complete DMEM [cDMEM]). Cells were plated on non-tissue cultured treated petri dishes (Corning) on day –8. Fresh medium was added on day –5 and on day –2 adherent cells were detached by trypsinization and collected. Unless specified otherwise, for acute/polarization studies, BMDMs were seeded in 12-well plates at $0.9×10^6$ BMDMs per well, and for training studies, BMDMs were seeded in 24-well plates at $0.2×10^6$ BMDMs per well.

On day –1 BMDMs were cultured with medium (naïve/untrained) or activated with 25 ng ml$^{-1}$ IFNγ, 10 ng ml$^{-1}$ LPS, 40 ng ml$^{-1}$ IL-4, 20 ng ml$^{-1}$ IL-13, 40 ng ml$^{-1}$ IL-10, 100 µg ml$^{-1}$ whole β-glucan particles (WGP Dispersible) – derived from the yeast *Saccharomyces cerevisiae* – or a combination of one or more activation stimuli, as specified in relevant figure legends. After 24 hr, the supernatant was replaced with fresh media. For acute activation/polarization studies, the BMDMs were left to rest for 2 hr before experiment. For training studies, the BMDMs were left to rest, fed on day 3 or 4 and experiments were carried out on day 6. The cDMEM used was supplemented with L929 cell line conditioned media, containing macrophage colony-stimulating factor (M-CSF) where two different batches were used. For all experimental repeats where β-glucan was used with or without the addition of IL-10, 20-, 20–10-, 7-, and 5% were used on days –8,–5, –2, 0 and 3/4 respectively (L929 batch gifted by Prof. Sheedy). For all other experiments 25-, 25-, 15-, 10-, and 7% were used on days –8,–5, –2, 0 and 3/4 respectively (batch gifted by Prof. Muñoz-Wolf).

To generate human monocyte-derived macrophages (MoDMs), mononuclear cells were isolated on day –7 from peripheral blood buffy coats obtained from the Irish Blood Transfusion Services (Dublin, Ireland) using density gradient centrifugation with Lymphoprep. Cells were cultured on plastic in RPMI media supplemented with 10% FBS (hereafter referred to as complete RPMI [cRPMI]) and 20 ng ml$^{-1}$ recombinant human M-CSF at $1×10^6$ cells ml$^{-1}$, seeded at $2×10^6$ cells per well. On day –4, 1 ml of media was carefully removed from each well and replaced with fresh cRPMI, supplemented with 20 ng ml$^{-1}$ recombinant human M-CSF. Non-adherent cells were removed by washing once on day –1 with pre-warmed PBS and incubated with cRPMI, supplemented with 5 ng ml$^{-1}$ M-CSF. After a minimum of 4 hr of rest, either more media was added (naïve/untrained) or the MoDMs were activated with 20 ng ml$^{-1}$ recombinant human IFNγ with 5 ng ml$^{-1}$ LPS or 20 ng ml$^{-1}$ IL-4 with 10 ng ml$^{-1}$ IL-13, the latter with or without 20 ng ml$^{-1}$ IL-10. After 24 hr (day 0), the media was removed and replaced with cRPMI, supplemented with 5 ng ml$^{-1}$ M-CSF. For acute activation/polarization studies, the MoDMs were left to rest for 2 hr before experiment. For training studies, the MoDMs were left to rest, fed on day 3 (media supplemented with 2.5 ng ml$^{-1}$ M-CSF) and experiments were carried out on day 6. Supply of human blood products from IBTS was approved by clinical indemnity to F. Sheedy.

## BCG infection

Mature BMDMs were seeded in 24-well plates at $0.5×10^6$ cells (acute activation) or $0.2×10^6$ cells (training) per well on day –2. For imaging of internalized BCG in untrained and trained BMDMs, $0.2×10^6$ BMDMs were seeded on circular glass coverslips (placed in 24 well plates, one coverslip per well) that had been previously treated with sodium hydroxide to aid attachment.

BMDMs were incubated with media (naïve) or activated as outlined above on day –1, for 24 hr before BMDMs were washed and fresh media was added. For acute activation studies, BMDMs were infected with BCG Denmark minimum 3 hr later. For training studies, BMDMs were fed on day 3 and infected with BCG Denmark on Day 6. At the time of infection, the media was removed and replaced with fresh media, either with or without (negative control) bacteria.

Regarding infection dose (BCG per cell), for infection in acutely activated BMDMs, cell number per well was assumed to be $0.5×10^6$. For infecting trained BMDMs, three extra wells were prepared for all conditions and BMDMs were removed from these wells by trypsinization on day 5, pooled and counted in triplicate. Prepared BCG single cell suspension (see below) was diluted to reach an intended infection dose of 5 BCG per cell (as measured by OD$_{600}$, where 0.1 is estimated to be $10×10^6$ bacteria ml$^{-1}$). Each infection dose was measured via CFU counts (see below) and was thus subsequently corrected to an actual infection dose of roughly 30 BCG per cell.

Three hours post infection the supernatant was removed and the infected BMDMs were washed twice with Dulbecco's Phosphate Buffered Saline (PBS) to remove extracellular bacteria, and fresh media was added.

To measure cytokine secretion at each time point (3-, 27-, and 51 hr post infection), supernatant was collected and filtered (polyethersulfone [PES] 0.22 μm Luer lock syringe filter; Millex-GP) to remove BCG, prior to specific cytokines being measured by ELISA.

For CFU counts, at each time point the media was removed, wells were washed once with PBS before the BMDMs were lysed with 0.05% Tween20 in water. The cell lysate was diluted in pre-warmed (37 °C) 7H9 media, and 50 μl of dilutions $10^{-2}$-$10^{-6}$ were spread onto enriched 7H11 agar plates by dotting. Agar plates were incubated at 37 °C for 6 weeks and colonies were counted once a week from week 2 and on. CFU counts 3 hr gave the multiplicity of infection (MOI). Killing was assessed by the difference in CFU counts following the 3 hr time point, see accompanying dataset for comprehensive calculations.

For imaging internalized BCG (wells containing glass coverslips), media was removed, BMDMs were washed with PBS and then fixed with 4% paraformaldehyde (in PBS) overnight. The paraformaldehyde was then removed and coverslips were stored in PBS.

## Stimulation experiments

Naive (media control) or activated BMDMs were incubated with media or secondary stimuli on day 0 (acute activation/polarization) or day 6 (training). Secondary stimuli: 150 μg ml$^{-1}$ gamma-irradiated whole cells of *Mycobacterium tuberculosis* strain H37Rv (concentration measured by OD$_{600}$, where 100 mg ml$^{-1}$ was 0.32), 35 ng ml$^{-1}$ PAM3CSK4, 25 ng ml$^{-1}$ LPS, 5 μg ml$^{-1}$ Poly I:C, 0.55 μg ml$^{-1}$ R848, 10 μg ml$^{-1}$ CpG.

For histone methylation inhibition: on day –2 BMDMs were incubated with MTA (final concentration 1 mM) 1 hr prior to media incubation or activation as outlined above.

For metabolic inhibition experiments: on day 6 BMDMs were pre-incubated with 1 mM 2-DG for 3 hr, 2 μM oligomycin or 10 μM BPTES 1 hr prior to media incubation or secondary stimulation.

For glucose depletion training experiment: untrained or trained BMDMs were incubated with either the regular high glucose cDMEM or glucose depleted cDMEM, with DMEM devoid of glucose (Gibco), from day –2 until incubation with media or secondary stimulation on Day 6 (fed day 3 with the same media as given on day –2). Note, the glucose depleted cDMEM included 8% v/v FBS and the same amount of L929 conditioned media as outlined previously, both of which provided a source of glucose.

Naïve (media control) or activated MoDMs were incubated with media or secondary stimuli on day 0 (acute activation/polarization) or day 6 (training). Secondary stimuli: 250 μg ml$^{-1}$ gamma-irradiated whole cells of *Mycobacterium tuberculosis* strain H37Rv, 100 ng ml$^{-1}$ PAM3CSK4.

## Seahorse

All real-time measurements of oxygen consumption rate (OCR) and extracellular acidification rate (ECAR) were measured by using Seahorse system: Seahorse XFe96 Analyzer (Agilent). The analysis used was a Mitochondrial Stress Test (using a standard Agilent Seahorse protocol).

On day –2, mature BMDMs were seeded in a 96-well Seahorse plate (Agilent): 100,000 cells per well for acute activation studies or 30,000 cells per well for training studies. BMDMs were activated on day –1 as outlined and analyzed on either day 0 (24 hr after activation) or day 6 (fed on day 3), with or without secondary stimulation on day 5 (training studies).

The day before analysis, a Seahorse Cartridge (Agilent) was incubated overnight at 37 °C (incubator devoid of CO$_2$). The following day, the sterile water was replaced with Calibration Fluid at pH 7.4 incubated as before for 90 min before the analysis.

An hour before the analysis, in the Seahorse plate regular cDMEM was replaced with Seahorse XF DMEM, supplemented with 10 mM glucose, 1 mM pyruvate and 2 mM L-glutamine. The BMDMs were then incubated for an hour at 37 °C (incubator devoid of CO$_2$).

Fifteen min before the analysis, the cartridge was removed and reagents/inhibitors were added to their respective ports: oligomycin (port A, final concentration 10 μM), FCCP (port B, final concentration 10 μM), rotenone and antimycin-A (port C, final concentrations 5 μM each).

The analysis was carried out according to the manufacturer's instructions. Recorded values less than zero were disregarded.

For training experiments, protein concentration was used to standardize the OCR and ECAR measurements. Supernatant was removed, cells were washed once in PBS and 10 μl of RIPA buffer

was added per well. After pipetting up and down and scraping, the buffer was collected and samples from the same condition were pooled. Of pooled solution, 20 µl was used to measure protein concentration, using the Pierce bicinchoninic acid (BCA) assay kit (Thermo Scientific) according to the manufacturer's instructions (microplate procedure). Absorbance was measured at 560 nm.

## Metabolomics (LC-MS) sample preparation

On day –2, BMDMs were seeded in 6-well plates: $1 \times 10^6$ cells per well and activated on day –1 with IL-4/13 as outlined or incubated with media. BMDMs were fed on day 3 and incubated with irradiated *M. tuberculosis* on Day 6 for 24 hr. Prior to metabolite extraction, cells were counted using a separate counting plate prepared in parallel and treated exactly like the experimental plate. Supernatant was removed and cells were washed once in PBS. After aspiration, the BMDMs were kept at –80 °C or on dry ice. Metabolites were extracted by adding chilled extraction buffer (500 µl/$1 \times 10^6$ cells), followed by scraping (carried out on dry ice). Buffer was transferred to chilled eppendorf tubes and shaken in a thermomixer at maximum speed (2000 rpm) for 15 min at 4 °C. Following centrifugation at maximum speed for 20 min, roughly 80% of supernatant was transferred into labelled LC-MS vials, taking care to avoid pellet and any solid debris.

## Western blot sample preparation

On day –2, BMDMs were seeded in 6-well plates: $1 \times 10^6$ cells per well and activated on day –1 with IL-4/13 or β-glucan as outlined, or incubated with media as an untrained control. BMDMs were fed on day 3 and on day 6, the cells were washed once with PBS, before being lysed with RIPA buffer.

## Flow cytometry

On day –2, $0.8 \times 10^6$ mature BMDMs were seeded on non-tissue cultured treated 35 mm petri dishes (Corning) and activated as previously outlined on Day –1. The naive or activated/trained BMDMs were harvested at two separate time points: day 0 (24 hours post activation for acute activation characterisation) or day 6 (fed on day 3), with or without secondary stimulation on day 5 as specified in figure legends (training experiments).

For analysis, BMDMs were placed on ice for 30 min before harvesting with PBS-EDTA (5 mM) solution by gently pipetting up and down and transferring to flow cytometry tubes. Cells were incubated with Fixable Viability Stain 510 at RT for 15 min. After washing with PBS, cells were stained with anti-mouse Fc block, 15 min prior staining with CD80-FITC, CD206-PE, F4/80-PerCP-Cy5.5, CD11b-APC-eFluor 780, MHC class II-eFluor 450 for an additional 30 min at 4 °C. The cells were washed with PBS and resuspended in flow cytometry buffer (1% FBS in PBS). Samples were acquired on a BD Canto II flow cytometer and the data was analysed by using FlowJo software.

## Method details

### BCG preparation and plating

Bacille Calmette-Guérin (BCG) Denmark ($OD_{600}$ 0.1) was incubated in 7H9 media at 37 °C with rotation until the $OD_{600}$ 0.5–0.8 (logarithmic growth phase) was reached.

Single cell suspension was prepared as follows: bacteria were pelleted by centrifugation and 7H9 media was removed. Bacteria were vortexed with glass beads before pre-warmed (37 °C) DMEM was added and bacteria were left to sediment for 5 min. Bacterial suspension was carefully collected (avoiding disruption of the pellet) and centrifuged again to pellet large clumps. Suspension was collected (avoiding the pellet). Suspension was passed through a 26 G needle 15 times to disaggregate bacterial clumps.

Bacterial concentration was estimated by $OD_{600}$ (where 0.1 was estimated as $10 \times 10^6$ bacteria ml$^{-1}$) and infection dose was confirmed by colony forming unit (CFU) counts.

7H11 agar plates (for CFU counts) were made up as follows: 10.5 g Middlebrook 7H11 powder and 2.5 ml glycerol was dissolved in distilled water to reach 500 ml and autoclaved. Once cooled sufficiently, 50 ml sterile-filtered (0.22 µm; SteriCup [Millipore]) albumin-dextrose-sodium chloride (ADN) enrichment was added. For 1 liter of ADN enrichment: 50 g fatty acid-free, heat-shocked bovine serum albumin, 8.5 g sodium chloride and 20 g glucose were dissolved in sterile water to reach 1 liter.

## Staining and imaging of internalized BCG

For staining the coverslips contained in 24-well plates, PBS was removed and 17 µM Hoechst 33342 and 7.5 U Phalloidin-Alexa Fluor 647 (in 1 ml PBS) was added to each well. After half an hour (in the dark), the solution was removed and the coverslips were washed with PBS. A total of 200 µl Modified Auramine-O stain was added for 2 min (in the dark), after which the stain was removed and the wells were washed with PBS. To quench any extracellular BCG, 200 µl Auramine-O Quencher de-colorizer was added for 2 min (in the dark) and then the quencher was removed and the coverslips were washed with PBS. The coverslips were then fixed onto glass slides: 2 µl Vectashield mounting media was used per coverslip and clear nail polish was applied around the edge and allowed to dry completely. The coverslips were stained and imaged the same day.

Images of all samples were obtained using a Leica SP8 confocal microscope, taken with x40 magnification oil objective. Z-stack images through the entire cell were obtained and representative images were taken from these stacks.

## Liquid chromatography coupled to mass spectrometry (LC-MS)

Extraction buffer for isolating metabolites: 50% LC-MS grade methanol, 30% LC-MS grade acetonitrile, 20% ultrapure water (internal filtration system), valine-d8 final concentration 5 µM.

Hydrophilic interaction chromatographic (HILIC) separation of metabolites was achieved using a Millipore Sequant ZIC-pHILIC analytical column (5 µm, 2.1×150 mm) equipped with a 2.1×20 mm guard column (both 5 mm particle size) with a binary solvent system. Solvent A was 20 mM ammonium carbonate, 0.05% ammonium hydroxide; Solvent B was acetonitrile. The column oven and autosampler tray were held at 40 °C and 4 °C, respectively. The chromatographic gradient was run at a flow rate of 0.200 mL/min as follows: 0–2 min: 80% B; 2–17 min: linear gradient from 80% B to 20% B; 17–17.1 min: linear gradient from 20% B to 80% B; 17.1–22.5 min: hold at 80% B. Samples were randomized and analysed with LC–MS in a blinded manner with an injection volume was 5 µl. Pooled samples were generated from an equal mixture of all individual samples and analysed interspersed at regular intervals within sample sequence as a quality control.

Metabolites were measured with a Thermo Scientific Q Exactive Hybrid Quadrupole-Orbitrap Mass spectrometer (HRMS) coupled to a Dionex Ultimate 3000 UHPLC. The mass spectrometer was operated in full-scan, polarity-switching mode, with the spray voltage set to +4.5 kV/–3.5 kV, the heated capillary held at 320 °C, and the auxiliary gas heater held at 280 °C. The sheath gas flow was set to 25 units, the auxiliary gas flow was set to 15 units, and the sweep gas flow was set to 0 unit. HRMS data acquisition was performed in a range of $m/z=70$–900, with the resolution set at 70,000, the AGC target at $1×10^6$, and the maximum injection time (Max IT) at 120ms. Metabolite identities were confirmed using two parameters: (1) precursor ion m/z was matched within 5 ppm of theoretical mass predicted by the chemical formula; (2) the retention time of metabolites was within 5% of the retention time of a purified standard run with the same chromatographic method. Chromatogram review and peak area integration were performed using the Thermo Fisher software Tracefinder 5.0 and the peak area for each detected metabolite was normalized against the total ion count (TIC) of that sample to correct any variations introduced from sample handling through instrument analysis. The normalized areas were used as variables for further statistical data analysis.

## Preparation of β-glucan stock

The β-glucan particles (WGP Dispersible) were weighed and dissolved in sterile water to yield 10–15 ml of 25 mg ml⁻¹. This solution was left at room temperature overnight (8–16 hr) before sonication with a 150VT ultra sonic homogenizer with a 5/32" microtip. Whilst on ice, the solution was sonicated for 5 min, at 50% power and 50% time pulse rate, while the tip was immersed roughly 5 mm below the surface of the liquid. The β-glucan particles were then pelleted by centrifugation (1000 G, 10 min, room temperature) and the water was removed by careful decanting and replaced with 0.2 M NaOH in water, at a volume to reach 25 mg ml⁻¹. After 20 min, the β-glucan was washed three times with sterile water, using the same pelleting, decanting and replacement of solvent conditions as described. Finally, two last washes were carried out to replace the sterile water with sterile PBS. Prior to each use, the stock was thoroughly vortexed.

## ELISA

For detection of secreted TNFα, IL-6, and IL-10 from BMDMs, supernatants were collected and cytokines quantified by ELISA according to manufacturers' instructions (R&D Systems, Biolegend of Invitrogen), except antibody and sample volumes were halved.

## Western blot

BMDMs were lysed with RIPA buffer and the whole-cell protein lysates were separated on pre-cast Bolt 12% Bis-Tris Plus gels (Invitrogen, NW00127BOX) and transferred to nitrocellulose membranes. Membranes were subsequently probed using the relevant primary (histone H3, H3K4me3, H3K27me3 and H3K9me2) and secondary antibodies (anti-IgG) and imaged using the Odyssey Fc Imager (Li-Cor). Images of blots were analysed with ImageJ to calculate the relative optical density (ROD = area/percentage). ROD was then standardized to the untrained/media control samples: adjusted ROD values.

## Nitric oxide (NO) secretion

NO concentrations were quantified by indirect measurement of nitrite ($NO_2^-$) via the Griess Reagent System kit (Promega). The assay was carried out as per manufacturer's instructions and absorbance of 560 nm was measured.

## Reverse transcription quantitative PCR (RT-qPCR)

RNA was isolated via High Pure RNA Isolation Kit (Roche) according to the manufacturer's instructions and RNA was eluted in 50 μl water. RNA (minimum 100 ng) was reverse transcribed into complementary DNA (cDNA) with an M-MLV reverse transcriptase, RNase H minus, point mutant, in reverse transcriptase buffer, mixed with dNTPs, random hexamer primers and ribonuclease inhibitor (RNAseOUT). Quantitative PCR was performed using KAPA SYBR FAST Rox low qPCR Kit Master Mix in accordance with the instructions provided by the manufacturer, using QuantStudio 3 System technology. Primers (Key Resources Table) were designed in Primer BLAST and/or were also checked in Primer BLAST for specificity to the gene of interest. Where possible, primers were chosen to cross an exon-exon junction.

RNA expression was normalized to the internal references β-actin and/or TATAbox-binding protein, from the corresponding sample ($Ct_{gene}–Ct_{reference}=\Delta Ct$). Furthermore, ΔCt from control samples were subtracted from the ΔCt of each sample ($\Delta Ct_{treatment}-\Delta Ct_{ctrl} = \Delta\Delta Ct_{treatment}$). Fold change was calculated as $2^{(-\Delta\Delta Ct)}$. These calculations were carried out using Microsoft Excel.

## Quantification and statistical analysis

Data were evaluated on either on Prism version 8 for Windows or via R for Metaboanalyst generated analysis. Differences between two independent groups were compared via unpaired Student's t-test. For BCG infection studies where two independent groups were compared at several time points, multiple t-test analysis was employed, with Holm-Sidak correction for multiple comparisons. Differences were considered significant at the values of * $p<0.05$, ** $p<0.01$, *** $p<0.001$ and **** $p<0.0001$.

## Data and code availability

All raw data, calculations, results from statistical analyses (P values and 95% confidence interval) is available in the accompanying dataset:

*Source data 1*. Macrophage Innate Training Induced by IL-4 and IL-13 Activation Enhances OXPHOS Driven Anti-Mycobacterial Responses Dataset. Raw data, calculations and results of statistical analyses for all included figures.

Unedited western blots (with accompanying ponceau stains for each blot) and a labelled figure is included in:

*Figure 2—figure supplement 3—source code 1* Unedited and labelled western blots. Unedited western blots for histone H3, H3K4me3, H3K27me3 and H3K9me2, accompanying ponceau stains and figure where size and wells are labelled.

R-history for the metabolomics analysis (carried out by MetaboAnalyst) can be found in: *Figure 3—source data 1*. MetaboAnalyst R-history.
Statistical analysis carried out by MetaboAnalyst. Raw data can be found in *Source data 1*.

## Acknowledgements

The authors thank Dr Gavin McManus for his assistance with the imaging studies.

M Lundahl was funded by a Trinity College Dublin postgraduate studentship and this work is supported by Science Foundation Ireland (SFI) Research Centre, Advanced Materials and BioEngineering Research (AMBER) under Grant number 12/RC/2278_P2 E, SFI under Grant number 12/IA/1421 and 19FFP/6484 (E. Lavelle) and SFI under Grant number 15/CDA/3310 (E. Scanlan). S Gordon and M Mitermite acknowledge funding from Wellcome Trust PhD Studentship 109166/Z/15 /A and SFI award 15/IA/3154.

## Additional information

### Funding

| Funder | Grant reference number | Author |
|---|---|---|
| Trinity College Dublin | | Mimmi LE Lundahl |
| Science Foundation Ireland | 12/IA/1421 | Ed C Lavelle |
| Science Foundation Ireland | 19FFP/6484 | Ed C Lavelle |
| Science Foundation Ireland | 15/CDA/3310 | Eoin M Scanlan |
| Advanced Materials and Bioengineering Research | 12/RC/2278_P2 E | Ed C Lavelle |
| Science Foundation Ireland | 15/IA/3154 | Stephen V Gordon |
| Wellcome Trust | 109166/Z/15/A | Morgane Mitermite |

The funders had no role in study design, data collection and interpretation, or the decision to submit the work for publication. For the purpose of Open Access, the authors have applied a CC BY public copyright license to any Author Accepted Manuscript version arising from this submission.

### Author contributions

Mimmi LE Lundahl, Conceptualization, Data curation, Formal analysis, Investigation, Methodology, Writing – original draft, Writing – review and editing; Morgane Mitermite, Niamh C Williams, Ming Yang, Filipa M Lebre, Bojan Stojkovic, Formal analysis, Investigation, Methodology, Writing – review and editing; Dylan Gerard Ryan, Data curation, Formal analysis, Investigation, Methodology, Writing – review and editing; Sarah Case, Roisin I Lynch, Eimear Lagan, Methodology; Aoife L Gorman, Resources, Formal analysis, Investigation, Methodology, Writing – review and editing; Adrian P Bracken, Frederick J Sheedy, Investigation; Christian Frezza, Data curation, Investigation, Methodology, Writing – review and editing; Eoin M Scanlan, Formal analysis, Supervision, Project administration, Writing – review and editing; Luke AJ O'Neill, Formal analysis, Investigation, Writing – review and editing; Stephen V Gordon, Conceptualization, Resources, Formal analysis, Investigation, Methodology, Writing – review and editing; Ed C Lavelle, Conceptualization, Supervision, Investigation, Methodology, Writing – original draft, Project administration, Writing – review and editing

### Author ORCIDs

Mimmi LE Lundahl http://orcid.org/0000-0003-3924-4072
Morgane Mitermite http://orcid.org/0000-0001-9169-2134
Ed C Lavelle http://orcid.org/0000-0002-3167-1080

## Ethics

Animals were maintained according to the regulations of the Health Products Regulatory Authority (HPRA). Animal studies were approved by the TCD Animal Research Ethics Committee (Ethical Approval Number 091210) and were performed under the appropriate license (AE191364/P079).

## Decision letter and Author response

Decision letter https://doi.org/10.7554/eLife.74690.sa1
Author response https://doi.org/10.7554/eLife.74690.sa2

# Additional files

## Supplementary files

• Transparent reporting form

• Source data 1. Macrophage Innate Training Induced by IL-4 and IL-13 Activation Enhances OXPHOS Driven Anti-Mycobacterial Responses Dataset. Raw data, calculations and results of statistical analyses for all included figures.

## Data availability

All data generated or analysed during this study are included in the manuscript and supporting files. Source data were submitted to Mendeley (https://doi.org/10.17632/ncbph43m85.3).

The following dataset was generated:

| Author(s) | Year | Dataset title | Dataset URL | Database and Identifier |
| --- | --- | --- | --- | --- |
| Lundahl MLE, Mitermite M, Ryan D, Case S, Williams N, Yany M, Lynch R, Lagan E, Lebre F, Gorman A, Stojkovic B, Bracken A, Frezza C, Sheedy F, Scanlan E, O'Neill L, Gordon S, Lavelle E | 2022 | Macrophage Innate Training Induced by IL-4 Activation Enhances OXPHOS Driven Anti-Mycobacterial Responses | https://doi.org/10.17632/ncbph43m85.3 | Mendeley Data, 10.17632/ncbph43m85.3 |

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
