## [Editor Report]

In this valuable study Lundahl et al., examine the role IL-4/13/10 cytokines have on bone marrow-derived macrophages and the modulation of trained immunity – currently, the mechanisms involved in trained immunity remain elusive. The authors demonstrate in a convincing fashion that six days following priming with IL-4/13 was associated with increased oxidative phosphorylation metabolism and enhanced killing of BCG – IL-10 can attenuate this effect. The evidence provided helps support the authors' hypothesis that alternatively activated macrophages help regulate mycobacterial infection. The main weakness in this study concerns epigenetic data to confirm the functional role of trained immunity, this will hopefully be answered by future studies.

---

## [Decision Letter]

**Decision letter after peer review:**

Thank you for submitting your article "Macrophage Innate Training Induced by IL-4 Activation Enhances OXPHOS Driven Anti-Mycobacterial Responses" for consideration by *eLife*. Your article has been reviewed by 3 peer reviewers, and the evaluation has been overseen by a Reviewing Editor and Satyajit Rath as the Senior Editor. The reviewers have opted to remain anonymous.

Essential revisions:

General Concerns

1. Several of the experimental data sets are limited to two experimental replicates (please see Figure 1, 3, 6, S1). These may be enough to calculate t-statistic, but there is no way to assess whether the assumption of normality is true and needless to say that if they employed a non-parametric test, it will not achieve p<0.05. These experiments require additional replicates.

2. It is unclear how "killed BCG" values were calculated, and the text and material and methods are not consistent. is killing and not just simply lack of mycobacterial replication. There could be a combination of BCG replication and killing and with this assay is very difficult to discriminate that. Does this represent the difference in CFU between 3h (i.e. after the initial inoculum) and each subsequent time point. Since the baseline is different, this calculation will exaggerate differences. The actual viable CFU at each time point should be presented on log scale, and separately as normalised ratios to the 3h data. If any plots presenting 'killing' in the current format are presented, then the method used to calculate killing needs to be clearly outlined in the methods or relevant figure legend. Furthermore, the colony counts are restricted to intracellular bacteria. Neither BCG or Mtb are obligate intracellular organisms. The authors should also assess extracellular bacteria, and quantify macrophage cell death over this time course, which may obviously release bacteria into the extracellular space.

Specific Concerns

3. Please clarify with regards to the potential specific effects of IL-4 and IL-13. The authors use a combination of IL-4/IL-13 in their experiments: what is the IL-4 and IL-13 effects separately? One could envisage that maybe only one of the cytokines induces these effects.

4. What is the effect of IL-4 or IL-13 on trained immunity induced by microbial stimuli such as β-glucan or BCG?

5. The metabolic studies of the IL-4/14 trained macrophages are very interesting. Is it correct to say blocking glutaminolysis before the secondary stimulation blocked cytokine production? How about if glutaminolysis would be blocked during the initial 24h exposure to IL-4/13 (and not during the secondary stimulation): is the trained immunity effect inhibited as well? This experiment would be required to argue that glutaminolysis inhibits trained immunity.

6. Does the glucose-depleted medium used for the experiments in Figure 4 contain pyruvate? Pyruvate is a common compound in some growth media, and it can be an alternative substrate for glucose.

7. Does IL-10 inhibit trained immunity induced by microbial compounds as well?

8. Consistent with previous reports, the authors observed that (IL-4/13) macrophages on Day 1 have increased OCR and ECAR values. However, a critical comparison with (IFNγ/LPS) macrophages would be at day 6 but as with the BCG infection studies, the M(IFNγ/LPS) showed reduced viability. This is an important experiment that should be done in a way that cell death is not induced, and seahorse studies are needed to validate the transcriptional data. Currently, it looks like the gene expression analysis was performed in dead cells.

9. Are M(4/13) and M(4/13/10) expressing classical alternative activation markers also at day 6 and following stimulation (either with Mtb or PAM3CSK4)? These data would be important to answer how much surface expression profiles might correlate (or not) with functional responses, such as the pro-inflammatory response or the metabolic profile observed on Day 6 for M(4/13).

10. The comparisons between IL4/13 and IFNγ/LPS primed BMDM at 6 days are lacking for some key assays (BCG killing and Seahorse metabolic measurements) because the authors say that IFNγ/LPS primed BMDM had reduced viability at this stage. I don't understand why this is case. It reduces the value of the observations in IL4/13 primed BMDM at 6 days, because we can't evaluate their specificity. In addition, the validity/interpretation of cytokine and transcriptomic analyses of day 6 IFNγ/LPS primed BMDM is a particular concern because this is likely to be confounded/biased by the reduced viability of the cells. These data should be excluded.

11. What is the combined priming with IFNγ and LPS is supposed to be modelling biologically. If the purpose is to model Th1 modulation of macrophage biology, then would IFNγ priming by itself as a more appropriate comparator to Th2 IL4/13 cytokines.

12. The authors propose that IL4/13 priming of mouse BMDM induces epigenetic changes responsible for the delayed pro-inflammatory phenotype. They need to show this experimentally, by methylation analyses and/or ATAC sequencing. The methylation inhibitor data which are presented in Figure S3C using the methylation inhibitor are too preliminary to draw meaningful conclusions, particularly as an important control has not been presented – unstimulated BMDMs incubated with the methylation inhibitor (to assess for on- or off-target effects on baseline cellular function).

13. An important consideration is that IL4 in particular can be a stimulus for macrophage proliferation in mouse cells. Therefore, the authors need to experimentally assess whether their 'delayed' phenotype is in fact associated with cell cycle/proliferation.

14. Please clarify the interpretation of the data from experiments using the metabolic inhibitor 2-DG. I understood the objective was to reverse the metabolic programming mediated by IL4/13 priming, in which case I would expect the IL10 response to increasing in the presence of 2-DG. The fact that 2-DG leads to reductions in all the cytokine responses only serves to show that cytokine production requires active metabolism. In addition, there is no control condition of unstimulated BMDMs incubated with the metabolic inhibitors, which would again be important to assess for on- or off-target effects on baseline cellular function.

15. The link between changes in metabolism and control of Mtb needs to be verified experimentally, particularly to test the hypothesis that inhibition of oxidative phosphorylation attenuates bacterial killing. Key experiments, at least the effects of +/-IL4/13 and +/-IFNγ priming on differential cytokine responses to restimulation after six days, and mycobacterial growth restriction need to be verified in human monocyte-derived macrophages.

16. The Extracellular flux studies comparing IL-4/IL-13 vs IL-4/IL-13/IL-10 training are missing and would be critical to support the cytokine and transcriptional profile (Figure 6). This will address an important point regarding the functional metabolic consequences of IL-10 during macrophage training, as claimed in the manuscript.

Other

Why the title only considers IL4 and not IL-13 the experiments are performed with both cytokines? The title should be revised.

Most of the evidence cited is derived exclusively from mouse models that have not been replicated in human macrophages. The provenance of data should be clearer throughout the introduction and this limitation of the evidence more prominently stated.

The authors acknowledge in the Introduction and Discussion that the concept of 'M1 and M2 macrophages' oversimplifies the complexity of macrophage plasticity, they still use the dichotomy as a framework for experimental design and employ the terminology throughout the manuscript. This makes for inconsistent reading and the manuscript would be improved if the experiments & results were contextualised within macrophage plasticity as a broad concept, with references to 'M1' and M2' removed.

Lines 61-63, the assertion that a metabolic shift to glycolysis is 'crucial for effective killing' of mycobacteria is overstated. The evidence is limited to associations rather than causation.

Lines 66-68, the strength of the assertion that it is established that Th1 priming of macrophages confers the ability to kill Mtb is overstated. This has never been reproducibly shown in human cells. The references given for this assertion do not contain data to support it and should be revised. In addition, one of the references provided for the subsequent assertion that alternative activation of macrophages opposes bactericidal responses and leads to enhanced bacterial burden (Orecchioni et al.,) does not contain any data about bacterial or mycobacterial control in this context. This should be clarified in case it is not the intended reference, or removed.

Lines 75-76, the description of innate training as 'non-specific' is misleading. More accurate to describe this as a mechanism for providing heterologous immunity.

Lines 79-80, the assertion that BCG vaccine protection against Mtb is mediated primarily by trained immunity is misleading. This has not been shown in man.

Lines 108-117 can be deleted, as they provide unnecessary repetition of the introduction.

Lines 147-148, the sentence doesn't make sense as written.

The data presented in figure 1B are referred to as measuring the 'multiplicity of infection' in the different conditions, however, in the text and methods, the authors state that they infect the BMDMs with 30 bacteria per cell, which would therefore be the MOI. Instead, the data presented in 1B appears to be a measure of differential bacterial uptake between the conditions. This should be clarified.

The cytokine ELISA results in Figure 1 (BCG), Figure 2 and S2 (irradiated Mtb) and Figure S3 (PCSK) have some inconsistencies which need to be clarified. The unprimed unstimulated BMDMs in Figure 1 do not secrete any TNF at baseline, however in some experiments within the subsequent figures, cells in the same unprimed/unstimulated condition secrete >500pg/ml TNF. This discrepancy should be noted and potential reasons for it commented upon. In addition, cytokine ELISA assays typically have a limit of detection which is higher than the 0pg/ml presented as the lower limit of the Y axes in these figures. The figures should be adapted to show this limit of detection. Finally, the presentation of this data is split across multiple core and supplementary figures, and different Y-axis scales are used for plots presenting data for the same cytokine (e.g. TNF in Figure S3a: Y-axis scale 0-2700; TNF in Figure S3b: Y-axis scale 0-15000). This hinders the readers' interpretation, particularly as the results are discussed within one paragraph (lines 173-184) with comparisons drawn therein by the authors. The data should be presented within one figure and consistent Y-axis scales used for each cytokine.

Figure 3A, the application of oligomycin, FCCP and rotenone and antimycin-A need to be explained in the legend, and in the results narrative.

The nitric oxide pathway data presented is preliminary, particularly as the authors state the BMDMs did not release NO in relation to their stimulus of interest (irradiated Mtb), so LPS is used instead. The rationale for switching stimulus to LPS is unclear. Other mycobacterial stimuli should instead be tested, e.g. live BCG or Mtb.

The reference in line 486 (Kahnert et al.,) is stated to show that "acute alternative macrophage activation has been demonstrated to lead to reduce control of *M. tuberculosis* growth". This is not shown in this paper, which does not contain any experiments assessing mycobacterial growth or killing. The reference should be altered or removed, or the preceding statement adapted.

It was assumed that the authors have limited bacterial viability assessments to BCG for lack of access to appropriate biosafety containment level 3 facilities. Ideally, these should be extended to Mtb to confirm that the effects are replicated in bacteria with the full complement of intracellular survival/ immune evasion mechanisms, and more directly relevant to disease. If they are not able to do that, this limitation of their data needs to be highlighted in the discussion.

[Editors' note: further revisions were suggested prior to acceptance, as described below.]

Thank you for resubmitting your work entitled "Macrophage Innate Training Induced by IL-4 and IL-13 Activation Enhances OXPHOS Driven Anti-Mycobacterial Responses" for further consideration by *eLife*. Your revised article has been evaluated by Satyajit Rath (Senior Editor) and a Reviewing Editor.

The manuscript has been improved but there are some remaining issues that need to be addressed, as outlined below:

The mechanisms remain to be fully resolved, in particular, the lack of epigenetic data to support their central conclusion that IL4/13 priming mediates the observed effects on macrophages by via trained immunity. Examination of classically activated macrophages (because IFNγ/LPS primed cells appear to die over 6 days). Finally, the lack of replication of key findings in human MDM and Mtb killing. For this reason, these limitations should be discussed in full.

---

## [Author Response]

Essential revisions:General Concerns1. Several of the experimental data sets are limited to two experimental replicates (please see Figure 1, 3, 6, S1). These may be enough to calculate t-statistic, but there is no way to assess whether the assumption of normality is true and needless to say that if they employed a non-parametric test, it will not achieve p<0.05. These experiments require additional replicates.

With regard to the BCG infection studies (Figure 1, Figure 6 and Figure S1), we have carried out additional repeats:

**Author response image 1. sa2fig1:** Fig 1 and Fig 1-figure supplement 1 (n = 2 on Day 0, n = 1 on Day 6).

**Author response image 2. sa2fig2:** Fig 6 and Fig 6-figure supplement 2 (n = 2 on Day 0, n = 1 on Day 6).

As can be seen in Author response image 1 and Author response image 2, these repeats support the presented data in the paper. This data has however not been added, as the CFU counts are on a different scale and skew the data to a point where consistent differences cannot be perceived.

For Figure 3, another rep has been added, the data added to the existing figure.

2. It is unclear how "killed BCG" values were calculated, and the text and material and methods are not consistent. is killing and not just simply lack of mycobacterial replication. There could be a combination of BCG replication and killing and with this assay is very difficult to discriminate that. Does this represent the difference in CFU between 3h (i.e. after the initial inoculum) and each subsequent time point. Since the baseline is different, this calculation will exaggerate differences. The actual viable CFU at each time point should be presented on log scale, and separately as normalised ratios to the 3h data. If any plots presenting 'killing' in the current format are presented, then the method used to calculate killing needs to be clearly outlined in the methods or relevant figure legend.

Since the reviewers pointed out that the method used to address mycobacterial killing, i.e. the difference in CFU following the 3-hour time point, was not clearly explained, this has been added where appropriate in the text, figure legends and methods. In the methods we have also referred to the accompanying dataset, where the CFU counts and calculations can be found.

Furthermore, the decision to not show CFU counts on a log scale was made to be able to see the differences in killing capacity between the different activation/training conditions. When presented on a log scale, these differences are not apparent.

Furthermore, the colony counts are restricted to intracellular bacteria. Neither BCG or Mtb are obligate intracellular organisms. The authors should also assess extracellular bacteria, and quantify macrophage cell death over this time course, which may obviously release bacteria into the extracellular space.

During optimisation of the BCG infection protocol, samples for CFU counts were taken from lysed cells (intracellular bacteria) and from the supernatant (extracellular bacteria). However, during these initial optimisation studies we saw no marked differences in extracellular bacteria between the untrained media control and the IL-4/13 trained BMDMs (see Author response image 3). Furthermore, for several optimisation studies/experimental reps, the CFU counts for the supernatant didn’t reach our cut-off of 30 colonies and thereby had to be excluded.

**Author response image 3. sa2fig3:** Following BCG infection on Day 6, where BMDMs were incubated with media control or IL-4/13 on Day -1 for 24 hours.

Furthermore, the potential issue of macrophage cell death (and subsequent release of internalized bacteria) was a factor of concern for these experiments, which was a primary reason for carrying out the confocal microscopy of BCG infected cells. There was no indication that the IL-4/13 activated cells were dying within the time span of these infection studies; representative images are shown (Figure 6E).

With these factors in mind, intracellular CFU, with the accompanying confocal microscopy, were deemed an accurate reflection of the intracellular bacterial survival for the experiments reported here.

Specific Concerns3. Please clarify with regards to the potential specific effects of IL-4 and IL-13. The authors use a combination of IL-4/IL-13 in their experiments: what is the IL-4 and IL-13 effects separately? One could envisage that maybe only one of the cytokines induces these effects.

To address the reviewer’s concern about the roles of IL-4 and IL-13 regarding training, we have added Figure 2—figure supplement 1E. This experiment showed how both cytokines could enhance inflammatory responses, yet that IL-4 and IL-13 together resulted in the greatest training effect, especially with regard to secondary stimulation with irradiated Mtb, therefore both were used for all subsequent experiments. Given that both cytokines are typically used together as M2 polarizing stimuli in murine studies, this decision matched prior work in the field. This experiment was carried out at an early stage of the project, but was not included initially due to the Mtb data – that training with IL-4 and IL-13 in this case did not enhance TNF, which did not concur with our prior work. Based on troubleshooting, we found that this was due to an aliquot of irradiated Mtb having been freeze-thawed too many times. However, due to the reduction of IL-10, we still observed a key component of the training profile and an overall skewing towards pro-inflammatory responses.

4. What is the effect of IL-4 or IL-13 on trained immunity induced by microbial stimuli such as β-glucan or BCG?

Although the reviewers raise a very interesting question regarding how these cytokines interact with other training stimuli, we think that adequately answering these questions are beyond the scope of this story. To adequately answer this question one would need to consider numerous proportions/ratios of the signals given. Furthermore, prior work by Mihai Netea has for instance highlighted that concentration matters hugely when it comes to training; as an example, a low concentration of LPS causes training, whereas a high concentration causes tolerance. In this case, when considering very different mechanisms at play – comparing IL-4/13 training with either BCG or β-glucan training – one would need to consider the strength of each stimulus, try how to provide each to an equal extent, or at least know which is more dominant in one experiment and vary it accordingly. With all these considerations in mind, we believe that this is an interesting question which we cannot adequately answer within the scope of this story.

5. The metabolic studies of the IL-4/14 trained macrophages are very interesting. Is it correct to say blocking glutaminolysis before the secondary stimulation blocked cytokine production? How about if glutaminolysis would be blocked during the initial 24h exposure to IL-4/13 (and not during the secondary stimulation): is the trained immunity effect inhibited as well? This experiment would be required to argue that glutaminolysis inhibits trained immunity.

We agree with this comment that the studies carried out does not answer whether glutaminolysis is important for the induction of training. However, the work herein does show that glutaminolysis is important for driving the enhancement of TNFα and IL-6 following secondary stimulation. This, along with other metabolic data, demonstrated a continued use of M2-typical metabolism. We did consider the suggestion above, but we reasoned that blocking glutaminolysis would impede the acute activation induced by IL-4 and IL-13. Given that previous work has outlined that an upregulation of glutaminolysis is a key component of IL-4/13 activation of macrophages; if glutaminolysis was blocked prior to IL-4/13 activation and there was an inhibition of training, we would not be able to isolate the role of glutaminolysis specifically, as it is a key component of acute alternative activation.

6. Does the glucose-depleted medium used for the experiments in Figure 4 contain pyruvate? Pyruvate is a common compound in some growth media, and it can be an alternative substrate for glucose.

The media does not include pyruvate, this will be amended in Materials and methods.

7. Does IL-10 inhibit trained immunity induced by microbial compounds as well?

This was a very interesting question and we have carried out new experiments, investigating how IL-10 affects training induced by β-glucan (see Figure 6G). These data indicate that IL-10 can also modulate training induced by other stimuli.

8. Consistent with previous reports, the authors observed that (IL-4/13) macrophages on Day 1 have increased OCR and ECAR values. However, a critical comparison with (IFNγ/LPS) macrophages would be at day 6 but as with the BCG infection studies, the M(IFNγ/LPS) showed reduced viability. This is an important experiment that should be done in a way that cell death is not induced, and seahorse studies are needed to validate the transcriptional data. Currently, it looks like the gene expression analysis was performed in dead cells.10. The comparisons between IL4/13 and IFNγ/LPS primed BMDM at 6 days are lacking for some key assays (BCG killing and Seahorse metabolic measurements) because the authors say that IFNγ/LPS primed BMDM had reduced viability at this stage. I don't understand why this is case. It reduces the value of the observations in IL4/13 primed BMDM at 6 days, because we can't evaluate their specificity. In addition, the validity/interpretation of cytokine and transcriptomic analyses of day 6 IFNγ/LPS primed BMDM is a particular concern because this is likely to be confounded/biased by the reduced viability of the cells. These data should be excluded.

By “reduced viability” what is meant is that following IFNγ and LPS stimulation, cell numbers clearly reduced over time, leading to difficulties when measuring extracellular flux via Seahorse (detection limit was not reached). Moreover, whilst stimulating these BMDMs with a relatively small amount of TLR ligand or killed bacteria clearly caused both transcriptional and cytokine secretion changes, indicating that there were enough BMDMs present for these experiments/analyses, however, when given live BCG bacteria, the IFNγ/LPS trained cells died within 24 hours. These comments have now been added to the text to aid clarity. Due to these observations, we believe that the IFNγ/LPS trained cells have reduced viability, which unfortunately leads to certain experimental procedures being impossible to do.

However, our opinion is that this additional group is not strictly necessary; showing the metabolic and bacterial killing profile on Day 0, is crucial to highlight how much the M(IL-4/13) change over time, and overall showing these differences by comparison to the untrained cells is a more important control.

9. Are M(4/13) and M(4/13/10) expressing classical alternative activation markers also at day 6 and following stimulation (either with Mtb or PAM3CSK4)? These data would be important to answer how much surface expression profiles might correlate (or not) with functional responses, such as the pro-inflammatory response or the metabolic profile observed on Day 6 for M(4/13).

Nos2, Arg1, CD80, MHC II and CD206 were included in the submitted form of the manuscript, so we have included Retnla and Chil3 +/- Mtb stimulation in Figure 6—figure supplement 1E.

11. What is the combined priming with IFNγ and LPS is supposed to be modelling biologically. If the purpose is to model Th1 modulation of macrophage biology, then would IFNγ priming by itself as a more appropriate comparator to Th2 IL4/13 cytokines.

The combination of IFNγ and LPS is the norm for inducing classical macrophage activation. The aim is to investigate these heavily researched macrophages profiles in the context of training and mycobacterial infection of the cells. Furthermore, in the context of an infection, PRR activation would occur in combination with IFNγ signalling and we therefore consider this model to not be irrelevant to this context. Moreover, in preliminary results with inducing M1 macrophage activation with IFNγ and LPS alone, we observed that the “typical” M1 macrophage profile was achieved with the two stimuli together. Crucially regarding iNOS (important for Mtb killing in murine models), where optimisation studies revealed that the addition of LPS was required to effectively enhance its transcription.

**Author response image 4. sa2fig4:** 100 ng/ml IFNγ, 50 ng/ml LPS, 50 ng/ml each. Nos2 expression after 24 hours.

When it came to optimising the BMDM innate training protocol, the concentrations of IFNγ and LPS needed to be revised, as using 50 ng/ml of each resulted in cell death within 72 hours of activation. Therefore, we reduced the concentrations to 25 ng/ml IFNγ and 10 ng/ml LPS, which allowed significant upregulation of classic M1 markers (including Nos2 expression) whilst resulting in enhanced viability.

12. The authors propose that IL4/13 priming of mouse BMDM induces epigenetic changes responsible for the delayed pro-inflammatory phenotype. They need to show this experimentally, by methylation analyses and/or ATAC sequencing. The methylation inhibitor data which are presented in Figure S3C using the methylation inhibitor are too preliminary to draw meaningful conclusions, particularly as an important control has not been presented – unstimulated BMDMs incubated with the methylation inhibitor (to assess for on- or off-target effects on baseline cellular function).

Whilst the role of epigenetics is certainly of interest for future studies, the primary emphasis of this manuscript is the role of metabolism. And with regard to metabolism we have used respiration assays, LC-MS, nutrient depletion and targeted disruption of metabolic pathways to support our findings. Again, whilst we agree that the role of epigenetics is of great importance and its implicated role is preliminary based on our data, a detailed examination of its role via ATAC sequencing is another dimension of this story that we think is outside the scope of this manuscript. To try to gather more preliminary data to aid this future research, we did carry out western blot analysis to see if we could identify which type of histone methylation was occurring, however, there were no marked differences in methylation of any of the sites checked (Figure 2—figure supplement 3B-D).

13. An important consideration is that IL4 in particular can be a stimulus for macrophage proliferation in mouse cells. Therefore, the authors need to experimentally assess whether their 'delayed' phenotype is in fact associated with cell cycle/proliferation.

Cell proliferation is most certainly an important factor to consider – in fact enhanced proliferation has been proposed to be a key component of training. To take this factor into account, prior to infection experiments, extra wells were plated, given the various conditions, and taken off with trypsin the day before infection and counted. The amount of live bacteria used and the results were adjusted accordingly so all are standardized to the same number of cells (this is stated in relevant figure legends and calculations are outlined in the accompanying dataset). Similarly, regarding extracellular flux assays, we used the BCA assay to measure protein content, to gain an estimate of cell number and values were adjusted accordingly. Calculations can be viewed in the accompanying dataset.

For measuring cytokine responses to various stimuli, we opted for measuring several cytokines to see the overall effect upon the cell response profile. The fact that we saw enhanced pro-inflammatory cytokine secretion specifically in the trained M(IL-4/13), whilst IL-10 was reduced, indicates a shift in profile which is not due to cell number.

14. Please clarify the interpretation of the data from experiments using the metabolic inhibitor 2-DG. I understood the objective was to reverse the metabolic programming mediated by IL4/13 priming, in which case I would expect the IL10 response to increasing in the presence of 2-DG. The fact that 2-DG leads to reductions in all the cytokine responses only serves to show that cytokine production requires active metabolism.

As mentioned in the text, although 2-DG is well-known as an inhibitor of glycolysis, it has also been demonstrated to also be an inhibitor of OXPHOS. As such, its use serves to show that cellular respiration is required to support cytokine responses. In this context, we were outlining which metabolic processes were key for the innate training phenotype induced by IL-4 and IL-13.

In addition, there is no control condition of unstimulated BMDMs incubated with the metabolic inhibitors, which would again be important to assess for on- or off-target effects on baseline cellular function.

We did measure cytokine secretion from BMDMs given no secondary stimulation for all tested conditions and no background cytokine secretion of IL-6 or IL-10 was detected, and in the case of TNF, there were no differences in background secretion between conditions (see Author response image 5). For these reasons this data was not included.

**Author response image 5. sa2fig5:** 

15. The link between changes in metabolism and control of Mtb needs to be verified experimentally, particularly to test the hypothesis that inhibition of oxidative phosphorylation attenuates bacterial killing. Key experiments, at least the effects of +/-IL4/13 and +/-IFNγ priming on differential cytokine responses to restimulation after six days, and mycobacterial growth restriction need to be verified in human monocyte-derived macrophages.

We did wish to carry out more infection experiments to investigate the role of metabolism further and elucidate whether this training phenomena were applicable in human macrophages. Unfortunately, due to complications caused by COVID-19, very limited infection experimentation was possible within a reasonable time frame. We were able to carry out some preliminary work with human monocyte-derived macrophages, which intimate that IL-4 and IL-13 activation does specifically enhance inflammatory responses to subsequent challenges, which is moreover reduced by the addition of IL-10, which is promising for future work in this field.

16. The Extracellular flux studies comparing IL-4/IL-13 vs IL-4/IL-13/IL-10 training are missing and would be critical to support the cytokine and transcriptional profile (Figure 6). This will address an important point regarding the functional metabolic consequences of IL-10 during macrophage training, as claimed in the manuscript.

We did look with seahorse with IL-4/13 with or without IL-10 after training, +/- PAM as a preliminary experiment (see Author response image 6), and we did not see any differences in their metabolic profiles. As such, we believe the mechanism of how IL-10 is impeding the training effect of IL-4/13 is independent of OXPHOS.

**Author response image 6. sa2fig6:** 

OtherWhy the title only considers IL4 and not IL-13 the experiments are performed with both cytokines? The title should be revised.

The choice of including only IL-4 in the title was done to make the title easier to read, however, we agree that it is more accurate to include both cytokines, so this has been amended.

Most of the evidence cited is derived exclusively from mouse models that have not been replicated in human macrophages. The provenance of data should be clearer throughout the introduction and this limitation of the evidence more prominently stated.

As stated in the text, the roles of iNOS and Arg1 for killing and impeding killing are as yet exclusive to the mouse model of TB. However, the key referenced papers referring to the role of glycolysis and effective killing of mycobacteria, such as Gleeson et al. 2016 and Hackett et al. 2020, highlight that this is the case not only in the mouse model but also for studies using human macrophages. As such, the discussed response profiles and the role of metabolism – the main focus of the article – is applicable for both murine and human studies.

The authors acknowledge in the Introduction and Discussion that the concept of 'M1 and M2 macrophages' oversimplifies the complexity of macrophage plasticity, they still use the dichotomy as a framework for experimental design and employ the terminology throughout the manuscript. This makes for inconsistent reading and the manuscript would be improved if the experiments & results were contextualised within macrophage plasticity as a broad concept, with references to 'M1' and M2' removed.

We agree with this reviewer's comment that the M1 and M2 labelling is less than ideal. However, as these macrophage states are defined with these labels in the literature, and we’re showing a profile that contradicts the stated dogma around M1 vs M2, we think using this as a framework makes it easier for a non-expert to follow the significance of our findings. On the other hand, this concern regarding the M1 vs M2 framework is the key reason we labelled our macrophages as M(4/13) etc – to define each activation profile by their stimuli, not by these labels.

Lines 61-63, the assertion that a metabolic shift to glycolysis is 'crucial for effective killing' of mycobacteria is overstated. The evidence is limited to associations rather than causation.

In light of this comment, the sentence has been changed to “has been postulated to be a key for effective killing”; we agree that this is more accurate.

Lines 66-68, the strength of the assertion that it is established that Th1 priming of macrophages confers the ability to kill Mtb is overstated. This has never been reproducibly shown in human cells. The references given for this assertion do not contain data to support it and should be revised. In addition, one of the references provided for the subsequent assertion that alternative activation of macrophages opposes bactericidal responses and leads to enhanced bacterial burden (Orecchioni et al.,) does not contain any data about bacterial or mycobacterial control in this context. This should be clarified in case it is not the intended reference, or removed.

We agree with the reviewer’s comment that the phrasing of this sentence does not comprehensively account for the current state of the art regarding IFNγ and subsequent TB protection. Therefore we have changed the sentence to “Due in part to the established ability of classically activated macrophages to kill *M. tuberculosis*, inducing Th1 immunity has been a key aim for TB vaccine development”, which more accurately describes the discussion and aim of the cited references. The use of past tense moreover still allows for the reader to understand the novelty of our findings in this text.

Regarding the Orecchioni et al. reference, we thank the reviewers for highlighting this error; this reference was accidentally moved during prior editing of the manuscript. This reference has now been removed.

Lines 75-76, the description of innate training as 'non-specific' is misleading. More accurate to describe this as a mechanism for providing heterologous immunity.

This sentence has been changed to “Unlike adaptive immune memory, the secondary challenge does not need to be related to the primary challenge.” to avoid being misleading.

Lines 79-80, the assertion that BCG vaccine protection against Mtb is mediated primarily by trained immunity is misleading. This has not been shown in man.

This sentence has been changed to “This ability of the BCG vaccine to train innate immunity is now believed to be a mechanism by which it induces its protection against *M. tuberculosis*.” By removing the word core, we hope to convey that it has been shown to be an additional mechanism, rather than its primary.

Lines 108-117 can be deleted, as they provide unnecessary repetition of the introduction.

Whilst this section is somewhat repetitive, and can be deleted, we decided to include this brief summary to remind the reader of the context and relevance of the experiments in that section. We thought it could also be useful in case someone skips/skims the introduction to know the key points before reading the results.

Lines 147-148, the sentence doesn't make sense as written.

This sentence has been changed to “Following BCG infection on Day 0 (Figure 1—figure supplement 1A), no IL-10 secretion was detected, and only bactericidal M(IFNγ/LPS) secreted detectable levels of TNFα.”

The data presented in figure 1B are referred to as measuring the 'multiplicity of infection' in the different conditions, however, in the text and methods, the authors state that they infect the BMDMs with 30 bacteria per cell, which would therefore be the MOI. Instead, the data presented in 1B appears to be a measure of differential bacterial uptake between the conditions. This should be clarified.

The experimental set-up was slightly different with regard to the BCG infection experiments: the media was removed completely and the BCG was added in fresh media. Similarly, in control wells, the media was changed, and thus the “0 h” came from this point to give the baseline. In other training experiments, the secondary stimuli were added on top of the media that was already present – that had been there for 48-72 hours – therefore the TNF had time to accumulate in the latter type of study. This clarification has been added to the methods section.

The cytokine ELISA results in Figure 1 (BCG), Figure 2 and S2 (irradiated Mtb) and Figure S3 (PCSK) have some inconsistencies which need to be clarified. The unprimed unstimulated BMDMs in Figure 1 do not secrete any TNF at baseline, however in some experiments within the subsequent figures, cells in the same unprimed/unstimulated condition secrete >500pg/ml TNF. This discrepancy should be noted and potential reasons for it commented upon.

The experimental set-up was slightly different with regard to the BCG infection experiments: the media was removed completely and the BCG was added in fresh media. Similarly, in control wells, the media was changed, and thus the “0 h” came from this point to give the baseline. In other training experiments, the secondary stimuli were added on top of the media that was already present – that had been there for 48-72 hours – therefore the TNF had time to accumulate in the latter type of study. This clarification has been added to the methods section.

In addition, cytokine ELISA assays typically have a limit of detection which is higher than the 0pg/ml presented as the lower limit of the Y axes in these figures. The figures should be adapted to show this limit of detection.

The graphs have been amended to acknowledge the actual lower detection limit in each case.

Finally, the presentation of this data is split across multiple core and supplementary figures, and different Y-axis scales are used for plots presenting data for the same cytokine (e.g. TNF in Figure S3a: Y-axis scale 0-2700; TNF in Figure S3b: Y-axis scale 0-15000). This hinders the readers' interpretation, particularly as the results are discussed within one paragraph (lines 173-184) with comparisons drawn therein by the authors. The data should be presented within one figure and consistent Y-axis scales used for each cytokine.

The data is presented this way to be able to clearly see differences between activation states at each time point. In essence, Figure 2—figure supplement 2A and B (formerly S3a and b) display data from two distinct situations; due to cell proliferation during the week the cell numbers are higher on day 6 and similarly at these two time points these are distinct activation states. Therefore, we chose to scale the graphs to see the differences between training/activation conditions at each time point.

Figure 3A, the application of oligomycin, FCCP and rotenone and antimycin-A need to be explained in the legend, and in the results narrative.

As requested, this has been added to the text and the figure legend.

The nitric oxide pathway data presented is preliminary, particularly as the authors state the BMDMs did not release NO in relation to their stimulus of interest (irradiated Mtb), so LPS is used instead. The rationale for switching stimulus to LPS is unclear. Other mycobacterial stimuli should instead be tested, e.g. live BCG or Mtb.

Regarding the secretion of NO, we did try to measure it after BCG infection, but none was detected. PAM was also tested (as a more relevant TLR ligand) but again, we had the same issue. We only saw consistently detectable levels of NO secretion with LPS, which is why it was included. We have seen that transcription of iNOS is enhanced by IL-4/13 training, and we wanted to include NO secretion, but as said it was difficult to detect with other stimuli.

The reference in line 486 (Kahnert et al.,) is stated to show that "acute alternative macrophage activation has been demonstrated to lead to reduce control of M. tuberculosis growth". This is not shown in this paper, which does not contain any experiments assessing mycobacterial growth or killing. The reference should be altered or removed, or the preceding statement adapted.

As highlighted by this reviewer’s comment, this sentence required re-phrasing to better account for the cited results.

The sentence has been changed to: “Whilst alternatively activated macrophages have been demonstrated to provide an intracellular environment that is favourable for TB growth (Kahnert et al., 2006), resulting in greater bacterial burden (Moreira-Teixeira et al., 2016) in an acute setting, there have been some conflicting data concerning the interplay between parasites and mycobacterial infections.”

It was assumed that the authors have limited bacterial viability assessments to BCG for lack of access to appropriate biosafety containment level 3 facilities. Ideally, these should be extended to Mtb to confirm that the effects are replicated in bacteria with the full complement of intracellular survival/ immune evasion mechanisms, and more directly relevant to disease. If they are not able to do that, this limitation of their data needs to be highlighted in the discussion.

We argue that the key finding in this paper is the unique phenotype of the IL-4 trained cells. BCG is used here as a model mycobacterium, as the switch to glycolysis has been implicated as important for effective mycobacterial killing. We have added more to the discussion to address this point and highlight the key finding of this paper: the unique macrophage phenotype counterintuitively induced by IL-4 and IL-13 training.

[Editors' note: further revisions were suggested prior to acceptance, as described below.]

The manuscript has been improved but there are some remaining issues that need to be addressed, as outlined below:The mechanisms remain to be fully resolved, in particular, the lack of epigenetic data to support their central conclusion that IL4/13 priming mediates the observed effects on macrophages by via trained immunity. Examination of classically activated macrophages (because IFNγ/LPS primed cells appear to die over 6 days). Finally, the lack of replication of key findings in human MDM and Mtb killing. For this reason, these limitations should be discussed in full.

We agree that there is more work required in future to fully elucidate the mechanism driving the IL-4 and IL-13-induced innate trained phenotype and to address how translational these findings are in the context of human macrophage responses during TB pathology. As such, we have further highlighted these points in the Discussion section of our manuscript. Furthermore, we highlight the challenge in certain experiments (crucially extracellular flux and BCG infection studies) to directly compare IL-4/IL-13 trained macrophages with IFNγ/LPS trained macrophages 6 days post activation. This is due to reduced viability of the latter cells and is now addressed in greater detail in the discussion, along with our hypothesized reason for this reduction in viability: that a macrophage robust metabolic shift to aerobic glycolysis is perhaps not sustainable long-term.